# Accelerating NNDescent with Random Projection Landmarks

## Abstract

NNDescent is a cornerstone algorithm for constructing approximate $k$-nearest neighbor graph, yet its efficiency is bottlenecked by the quality of its initial candidate graph. We introduce Random Projection Landmarks (RPL), an initialization strategy that leverages auxiliary random vectors to generate high-quality candidate sets via extreme projection orderings. Under mild conditions, we show that RPL identifies $c$-approximate nearest neighbors with high probability and, conditioned on discovering such an approximate neighbor, increases the probability of discovering a closer neighbor through neighbor-of-neighbor exploration. Our analysis provides a probabilistic explanation for faster convergence in early NNDescent iterations. Empirical evaluations on million-scale datasets demonstrate that RPL reduces total distance evaluations and speeds up convergence while maintaining competitive graph quality.

## 1. Introduction

The $k$-nearest neighbor ($k$NN) graph serves as a fundamental data structure in many data mining and machine learning tasks, such as outlier detection (Goodge et al., 2022), recommendation systems (Park et al., 2015), clustering tasks (Alshammari et al., 2021) and visualization tasks (McInnes et al., 2018). Due to the curse of dimensionality, constructing exact $k$NN graph in high-dimension is computationally prohibitive, typically requiring $O(n^2)$ time for a dataset of $n$ points (Andoni et al., 2018).

A common strategy is to view graph construction as issuing $n$ approximate $k$NN queries, where each data point acts as a query. Once the index is built, queries can be answered efficiently through its search structure. High-performance approximate nearest neighbor search (ANNS) systems such as Faiss (Douze et al., 2024), HNSW (Malkov & Yashunin,

[1]Anonymous Institution, Anonymous City, Anonymous Region, Anonymous Country. Correspondence to: Anonymous Author <anon.email@domain.com>.

Preliminary work. Under review by the International Conference on Machine Learning (ICML). Do not distribute.

2020), and ScaNN (Guo et al., 2020) support fast queries for out-of-sample vectors. However, when constructing a $k$NN graph on a fixed dataset, the expensive indexing phase will be the bottleneck for the $k$NN graph construction.

Other work on approximate $k$NN graph construction has explored divide-and-conquer strategies (Chen et al., 2009; Wang et al., 2012; Zhang et al., 2013), where the dataset is (recursively) divided into smaller subsets and local neighborhood graphs are constructed within each subset. These local graphs are then merged using either overlapping partitions or auxiliary subsets to form the global graph. Such divide-and-conquer strategies are often hindered by the "boundary problem," in which true nearest neighbors are separated into disjoint partitions. This necessitates either computationally exhaustive cross-joins to restore connectivity or results in a significant sacrifice in the final graph's recall.

The seminal work to construct approximate $k$NN graph is *NNDescent* (Dong et al., 2011), which dynamically refines the neighbor list through iterative updates on a random initial graph. Inspired by the principle that a "*neighbor of a neighbor is likely to be a neighbor*," NNDescent starts from an initial random graph, and each node explores the neighbors of its own neighbors to identify closer candidates. By performing these "local joins" and updating neighbor lists with the closest points found, the algorithm progressively refines the graph until it converges to a state where no further distance reductions are possible. However, the convergence efficiency of NNDescent's local search is highly sensitive to the quality of the initial neighbor selection (McInnes, 2021). In the presence of highly skewed datasets, a poor initial graph can lead to protracted convergence times or cause the algorithm to stagnate in local optima.

**Contribution.** We propose a theoretically motivated acceleration of NNDescent under cosine distance using *random projection landmarks* (RPL). Our method uses random directions to induce coarse angular orderings of the data, from which we derive landmark-based neighbor candidates. These landmarks bias the initial $k$NN graph and the subsequent neighbor-of-neighbor exploration toward points with high angular similarity, significantly reducing the number of uninformative distance evaluations.

Unlike uniform random initialization, RPL preserves angular locality in expectation and provides a geometry-aware

starting graph for NNDescent. We show that, under mild assumptions, points that are close in cosine distance co-occur within the same or neighboring landmark-induced buckets with high probability, yielding an initial $k$NN graph that contains a non-trivial fraction of true nearest neighbors. This property ensures that the neighbor-of-neighbor exploration in NNDescent starts from a graph with higher local connectivity than uniform random initialization. Empirically, this leads to faster convergence and lower total computational cost, while maintaining higher final recall. RPL is also lightweight and complementary to existing NNDescent optimizations and libraries.

## 2. Preliminary and Related Work

We review NNDescent (Dong et al., 2011) and recent graph initialization enhancements, while establishing the geometric properties of random projections that provide the theoretical groundwork for this work.

### 2.1. NNDescent

Let $X = \{x_1, \ldots, x_n\} \subset \mathbb{R}^d$ be a dataset and let $d(\cdot, \cdot)$ be a distance measure. For a point $x \in X$, denote by $\mathrm{NN}_k(x)$ its $k$ nearest neighbors in $X$. The goal of approximate $k$NN graph construction is to build a directed graph $G = (V, E)$ with $V = X$ such that each node $x$ has outgoing edges to a set $N_k(x)$ approximating $\mathrm{NN}_k(x)$.

**Neighbor-of-neighbor heuristic.** NNDescent is an *iterative* algorithm based on the empirical observation that *neighbors of a point are likely to share neighbors*. Formally, if $y \in \mathrm{NN}_k(x)$ and $z \in \mathrm{NN}_k(y)$, then $z$ has a higher probability of belonging to $\mathrm{NN}_k(x)$ than a uniformly random point from $X$. This heuristic allows NNDescent to explore the local neighborhood structure of the data without exhaustive pairwise comparisons.

**NNDescent algorithm.** Algorithm 1 shows a simple implementation of NNDescent, starting from an initial random $k$NN graph. At each iteration, for every point $x$, the algorithm considers the approximate $k$NN $B(x)$. For each pair of points $(u, v) \in B(x)$, the distance $d(u, v)$ is evaluated. If $v$ is closer to $u$ than the farthest point in $N_k(u)$, the neighbor list of $u$ is updated accordingly. After all updates, each neighbor list is pruned to retain only the top-$k$ closest points. This process is repeated until convergence or for a fixed number of iterations.

**Computational complexity.** Let $T$ denote the number of NNDescent iterations. In each iteration, we run a local join on the approximate $k$NN of each point, leading to $O(k^2)$ distance computation. Thus, the overall time complexity is $O\left(T \cdot n \cdot k^2 \cdot c_d\right)$, where $c_d$ is the cost of a single distance computation (e.g., $O(d)$ for dense vectors with cosine distance). In practice, $k$ and $T$ are small constants, making

---

**Algorithm 1** NNDescent with random initialization

---

**Require:** For each $x$, $B(x) \leftarrow k$ random points from $X$
**Ensure:** For each $x \in X$, return $B(x)$ as the approximate $k$NN of $x$
1: **while** converged **do**
2:     **for** each $x \in X$ **do**
3:         **// Local join**
4:         **for** each pair $u, v \in B(x)$ **do**
5:             $dist \leftarrow d(u, v)$
6:             Update $B(u)$ with $(v, dist)$
7:             Update $B(v)$ with $(u, dist)$
8:         **end for**
9:     **end for**
10: **end while**
11: **Return** $B(x)$ for each $x \in X$

---

NNDescent near-linear in $n$ and scalable to large datasets. The space complexity is $O(nk)$ for storing the approximate $k$NN graph.

**Limitation.** NNDescent's effectiveness strongly depends on the initial graph. Under random initialization, the early neighbor lists contain little meaningful structure, and true neighbors are discovered only by chance. On highly skewed datasets, the algorithm may temporarily form poorly connected local clusters, making it difficult for true nearest neighbors across distant regions to be discovered.

**PyNNDescent (McInnes, 2021).** To handle the performance sensitivity of NNDescent, PyNNDescent library initializes the graph with random projection trees (RPT). The RPT-based initialization is inspired by the work (Dasgupta & Sinha, 2015; Bernhardsson, 2018) that shows an ensemble of several RPT potentially improves the accuracy of ANNS.

PyNNDescent implements RPTs that use the data point itself as the landmark to recursively partition the data. Points fall into the same leaf if they are close to several landmark points associated with internal nodes. PyNNDescent executes a local join within each leaf to construct a set of disjoint subgraphs, which are subsequently unified into a global initial graph. Though this structured initialization provides a high-quality starting point for NNDescent, it is challenging to characterize its convergence due to the data-dependent landmark selection. Another hurdle of data-dependent RPT initialization is the convergence to a local optimum on highly clustered data sets, as data-dependent landmarks tend to be selected from these clusters.

### 2.2. Extreme Random Projections

We first establish the theoretical properties of random projections that underwrite our landmark-based initialization for NNDescent. The proofs are provided in the appendix.

**Extreme random projections as directional proxies.** Let $x, y \in \mathbb{S}^{d-1}$ be fixed unit vectors, and let

$$r_1, \ldots, r_D \overset{\text{i.i.d.}}{\sim} \mathcal{N}(0, I_d)$$

be independent standard Gaussian random vectors. Define

$$r_* = \arg \max_{1 \leq \ell \leq D} \langle x, r_\ell \rangle$$

be the random vector closest to $x$. We study the asymptotic behavior of $\langle x, r_* \rangle$ and $\langle y, r_* \rangle$ as $D \to \infty$.

**Lemma 2.1** (Extreme Projection Decomposition). *Let $\rho = \langle x, y \rangle \in [-1, 1]$. Then, as $D \to \infty$,*

$$\langle x, r_* \rangle = \sqrt{2 \log D} + o_p(\sqrt{\log D}) ,$$
$$\langle y, r_* \rangle = \rho \sqrt{2 \log D} + \sqrt{1 - \rho^2} \, Z + o_p(\sqrt{\log D}),$$

*where $Z \sim \mathcal{N}(0, 1)$.*

We say $X_D = o_p(\sqrt{\log D})$ when $X_D$ grows strictly slower than $\sqrt{\log D}$ as $D \to \infty$. The proof in the appendix is based on the classical extreme-value for Gaussian maxima.

Lemma 2.1 shows that conditioning on maximal alignment with $x$ causes the selected random vector $r_*$ to behave as a noisy proxy for $x$. The signal term $\rho \sqrt{2 \log D}$ diverges for any $\rho > 0$, while the noise term remains $O(1)$, implying that rankings induced by $r_*$ increasingly reflect angular similarity to $x$ as $D$ grows.

**Order statistics of dependent Gaussian variables.** Let $Z_1, \ldots, Z_n$ be real-valued random variables such that each $Z_i \sim \mathcal{N}(0, 1)$ marginally, but the variables may be arbitrarily dependent. We denote the order statistics below

$$Z_{(1)} \geq Z_{(2)} \geq \cdots \geq Z_{(n)} .$$

**Lemma 2.2** (Expectation of the $m$-th Largest Gaussian). *For any $1 \leq m \leq n$,*

$$\mathbb{E}[Z_{(m)}] \leq \sqrt{2 \log(n/m)} + O(1).$$

## 3. NNDescent with Random Projection Landmarks on Cosine Distance

The proposed Random Projection Landmark (RPL) algorithm serves as an efficient initialization phase for NNDescent using cosine distance. It replaces traditional random initialization with a simple two-stage process, as shown in Algorithm 2.

First, it maps each data point to its $i$ most similar "landmark" vectors from a set of $D$ random vectors, effectively performing a soft-clustering based on angular proximity. Second, within each landmark's bucket, the algorithm prunes candidates to the $m$ closest points before performing a local

---

**Algorithm 2** *RPL*: Random projection landmarks

**Require:** $X \subset \mathbb{S}^{d-1}$, $D$ random vectors $r$, $m$, $i$
**Ensure:** For each $x \in X$, return $B(x)$ as the approximate $k$NN of $x$
1: Initialize $B(x) \leftarrow \emptyset$ for each $x \in X$
2: **// Multiple landmark assignment**
3: **for** each $x \in X$ **do**
4:     Compute top-$i$ closest vectors $r$, and insert into the $i$ buckets associated to these $i$ landmarks
5: **end for**
6: **// Landmark-induced local join**
7: **for** each bucket $B_r$ associated to the landmark $r$ **do**
8:     Keep the top-$m$ closest points to $r$
9:     **for** each pair $u, v \in B_r$ **do**
10:         $\rho \leftarrow \langle u, v \rangle$
11:         Update $B(u)$ with $(v, \rho)$
12:         Update $B(v)$ with $(u, \rho)$
13:     **end for**
14: **end for**
15: **Return** $B(x)$ for each $x \in X$

---

join. This ensures that the computational complexity of RPL remains bounded while producing a $c$-approximate $k$NN graph for a constant $c > 1$, which significantly reduces the search space for subsequent NNDescent iterations.

In spaces with a fixed doubling dimension, this landmark-based approach ensures that initial edges act as high-quality "bridges," allowing the global $k$NN structure to be recovered with minimal refinement.

The process of RPL can be viewed as constructing a "hash table" of $D$ buckets where each random vector landmark serves as a bucket representative. To further boost accuracy, we repeat the RPL procedure $L$ times and refine the output, analogous to constructing $L$ independent hash tables in ANNS (Andoni et al., 2015; Pham & Liu, 2022).

**Time complexity.** Since each point is represented by $i = O(1)$ landmark vectors, each landmark is associated with $O(m)$ points, and we repeat RPL $L$ times, the time complexity of RPL is $O(dnDL + dm^2 DL)$ with cosine distance. As we will use Fast Hadamard Transform (FHT) (Andoni et al., 2015) to simulate Gaussian random projections, the local join will dominate the runtime in practice.

## 4. Theoretical Analysis of RPL

We provide a theoretical analysis of how RPL accelerates the convergence of NNDescent. First, we establish the conditions under which the top-$m$ points in each bucket guarantee a $c$-approximate nearest neighbor for every data point (Theorem 4.2). We then quantify the probability of discovering *closer* neighbors during the subsequent NNDes-

cent iterations (Theorem 4.4). Finally, we characterize the overall improvement in the convergence rate achieved by this initialization (Theorem 4.6).

### 4.1. Top-$m$ Inclusion under Extreme Random Projections

Let $X = \{x_1, \ldots, x_n\} \subset \mathbb{S}^{d-1}$ be the dataset and $x \in \mathbb{S}^{d-1}$ be a fixed query point. We define the random vector closest to $x$ among $D$ random vectors:

$$r_* = \arg \max_{1 \leq \ell \leq D} \langle x, r_i \rangle.$$

For a data point $y$, define $\rho = \langle x, y \rangle$, and assume that $\rho < 1$. Applying Lemma 2.1 and 2.2, we have

**Lemma 4.1** (Top-$m$ Inclusion Probability). *As $D \to \infty$, the probability that $y$, s.t. $\langle x, y \rangle = \rho < 1$, lies among the top-$m$ data points ranked by $\langle r_*, \cdot \rangle$ satisfies*

$$\Pr[y \in \text{Top-}m(r_*)] =$$
$$\Pr\left[ Z \geq \frac{\sqrt{2 \log(n/m)} - \rho \sqrt{2 \log D}}{\sqrt{1 - \rho^2}} \right] + o(1),$$

*where $Z \sim \mathcal{N}(0, 1)$.*

We now show how RPL returns a $c$-approximate nearest neighbor for a query point on cosine distance.

**Theorem 4.2** (Random-Projection Landmark Inclusion Probability). *Let $x \in \mathbb{S}^{d-1}$ be a fixed query point and let $y \in X = \{x_1, \ldots, x_n\} \subset \mathbb{S}^{d-1}$ be data points with $\rho = \langle x, y \rangle \in (0, 1)$. Let $r_1, \ldots, r_D \sim \mathcal{N}(0, I_d)$ be i.i.d. Gaussian random vectors, with $D = n/m$, and define*

$$r_* = \arg \max_{1 \leq \ell \leq D} \langle x, r_\ell \rangle.$$

*Let $\text{Top-}m(r_*)$ denote the $m$ points in $X$ with largest inner products with $r_*$.*

*Then, as $D \to \infty$, up to universal constants,*

$$\Pr[y \in \text{Top-}m(r_*)] = D^{-\frac{1-\rho}{1+\rho}}.$$

*For any collection of points $\{y_i\}$ in $X$ with $\langle x, y_i \rangle \geq \rho$, we assume that $\langle y_i, y_j \rangle \geq \langle x, y_i \rangle \cdot \langle x, y_j \rangle$.*

*Then, the probability that at least one $y_i$ appears in $\text{Top-}m(r_*)$ is lower bounded by*

$$1 - \left( 1 - D^{-\frac{1-\rho}{1+\rho}} \right)^{|\{y_i\}|},$$

*where dependence of among projections $\langle y_i, r_* \rangle$ can only increase this probability.*

The triangle inequality for angular distance gives

$$\langle y_i, y_j \rangle \geq \langle x, y_i \rangle \langle x, y_j \rangle - \sqrt{1 - \langle x, y_i \rangle^2} \sqrt{1 - \langle x, y_j \rangle^2}.$$

As points become more concentrated within a narrow spherical cap, the error term in the angular triangle inequality vanishes. Therefore, our assumption that $\langle y_i, y_j \rangle \geq \langle x, y_i \rangle \cdot \langle x, y_j \rangle$ is a natural and reliable approximation in high-similarity regimes.

*Remark 4.3.* Theorem 4.2 indicates that, for a constant $c > 1$, we can find a $c$-approximate nearest neighbor for any fixed point $x \in X$ with probability at least $p > 0$. Then, by running RPL independently $L = O(\log n)$ times, and selecting the best result, the algorithm returns a $c$-approximate answer for all points in $X$ with probability at least $1 - n^{-\Omega(1)}$.

### 4.2. Convergence of NNDescent in Doubling Spaces

Let $y$ be a $c$-approximate nearest neighbor of $x$ and $y^*$ be the nearest neighbor of $x$, and define $d(x, y) \leq cR, d(x, y^*) = R$. We define the improvement set

$$\mathcal{Z}(x, y) = \{z \in X : d(x, z) < d(x, y)\}.$$

The following theorem justifies that if the improvement set $|\mathcal{Z}(x, y)|$ is large, given the starting point $y$, we will find an improvement point $z \in \mathcal{Z}(x, y)$ with high probability. This is because $y$ and $z$ tend to appear in the top-$m$ points associated to a random vector $r$.

**Theorem 4.4** (Random-Projection Bridge Discovery). *Let $x \in \mathbb{S}^{d-1}$ be a query point and let $y \in X$ be a discovered neighbor with $\langle x, y \rangle = \rho$. Define the improvement set*

$$\mathcal{Z}(x, y) = \{z \in X : \langle x, z \rangle > \rho\}.$$

*Let $r \sim \mathcal{N}(0, I_d)$ be a random projection vector, and let $\text{Top-}m(r)$ denote the $m$ points in $X$ with largest projection scores $\langle r, \cdot \rangle$.*

*Assume:*

*(A1) $y \in \text{Top-}m(r)$.*

*(A2) There exists $\gamma > 0$ such that for all $z \in \mathcal{Z}(x, y)$,*

$$\langle y, z \rangle \geq \gamma.$$

*Then there exists a constant $c(\gamma) > 0$ such that*

$$\Pr[\exists z \in \mathcal{Z}(x, y) \text{ s.t. } z \in \text{Top-}m(r) \mid y \in \text{Top-}m(r)] \geq$$
$$1 - \exp\left( -c(\gamma) |\mathcal{Z}(x, y)| \right).$$

*In particular, the probability of discovering a strictly better neighbor increases monotonically with the number of improving neighbors.*

Since we will show the convergence of NNDescent in doubling spaces, we present the formal definition here.

**Definition 4.5** (Doubling Dimension). A metric space $(X, d(\cdot, \cdot))$ has doubling dimension $d_0$ if every ball $B(x, cR)$ can be covered by at most $\alpha \cdot c^{d_0}$ balls of radius $R$ for a constant $c > 1$ and a constant $\alpha$.

We now show the convergence of NNDescent for $k = 1$.

**Theorem 4.6** (NNDescent Convergence with Random-Projection Landmarks)**.** *Let $(X, d(\cdot, \cdot))$ be a metric space with doubling dimension $d_0$. Fix a point $x \in X$, and suppose that the random-projection landmark initialization has discovered a $c$-approximate nearest neighbor $y$ of $x$, i.e.,*

$$d(x, y) \leq c \cdot d(x, y^*),$$

*where $y^*$ is the true nearest neighbor of $x$.*

*Then there exists a constant $\eta(d_0) > 0$ such that, in the next NNDescent iteration,*

$$\Pr[\exists z \in X \text{ with } d(x, z) < d(x, y)] \ \geq \ 1 - \exp\big(-\eta(d_0) \log c\big).$$

*Consequently, the approximation factor decreases geometrically, and NNDescent converges to exact nearest neighbors in*

$$O(d_0 \log c)$$

*iterations.*

*Proof.* Let $d(x, y^*) = R$, and $d(x, y) = cR$ for some $c > 1$. We process the proof with 3 steps.

**Step 1: Density of the Improvement Set.** Given the improvement set $\mathcal{Z}(x, y) = \{z \in X : d(x, z) < d(x, y)\}$, by the definition of the doubling dimension $d_0$, the ball $B(x, cR)$ can be covered by at most $M \leq \alpha c^{d_0}$ balls of radius $R$, where $\alpha$ is a constant. Since $B(x, R) \subseteq \mathcal{Z}(x, y)$, and $B(x, R)$ contains the local density $n_x(R)$ of the nearest neighbor, the "relative density" of improving points within the search radius of $y$ is bounded by:

$$\frac{|\mathcal{Z}(x, y)|}{|B(x, cR)|} \geq \frac{n_x(R)}{\alpha c^{d_0} n_x(R)} = \Omega(c^{-d_0}).$$

This implies that a significant population of points exists that are strictly closer to $x$ than $y$. As NNDescent explores $c$-approximate neighbors of $y$, we also have

$$\frac{|\mathcal{Z}(x, y)|}{|B(y, (c+1)R)|} \geq \frac{|B(x, R)|}{|B(x, (2c+1)R)|} = \Omega(c^{-d_0}) \ .$$

Therefore, the relative density of these improving points within $y$'s neighborhood remains $\Omega(c^{-d_0})$.

**Step 2: Bridge discovery probability.** Conditioning on the event that $y$ is discovered through a random-projection landmark induces positive alignment with high probability,

$$\langle y, z \rangle \geq \gamma \quad \text{for all } z \in \mathcal{Z}(x, y),$$

for some constant $\gamma > 0$ depending only on geometry.

From Theorem 4.4, the probability that at least one $z \in \mathcal{Z}(x, y)$ is discovered in the next NNDescent iteration is

$$1 - \exp\big(-\Theta(|\mathcal{Z}(x, y)|)\big) \ \geq \ 1 - \exp\big(-\Theta(c^{-d_0})\big).$$

**Step 3: Convergence via Local Density.** Once the initial "bridge" is crossed, the landmark's role is succeeded by the local density of the doubling space. For any current neighbor $z$ at distance $d(x, z) = c'R$, where $c' < c$, the doubling property ensures that the local neighborhood $B(z, \text{poly}(c')R)$ is sufficiently populated with points from the next improvement set $\mathcal{Z}(x, z)$. Specifically, the probability of finding a closer point $z'$ through a local join (exploring neighbors of neighbors) remains $\Omega((c')^{-d_0})$. This ensures a consistent contraction of the approximation factor. Because each successful iteration reduces the distance to $x$ by a factor $(1 - \epsilon)$ where $\epsilon$ is inversely proportional to $d_0$, the total number of iterations required to reach $c = 1$ is bounded by $O(d_0 \log c)$. □

*Remark* 4.7. RPL ensures that even if the initial neighbor $y$ is far from $x$ ($c$ is large), it is already directionally aligned with $x$ through a shared landmark. This pre-existing alignment provides a "high-speed corridor" that allows the very first NNDescent iteration to achieve a significant leap into the improvement set $\mathcal{Z}(x, y)$. By replacing the "hope" for random shortcuts with a geometric guarantee of connectivity, the bridge mechanism allows the algorithm to enter the $O(d_0 \log c)$ contraction phase immediately, significantly reducing the total number of distance computations required for convergence.

### 4.3. Data-independent vs Data-dependent Landmarks

A fundamental distinction between clustering-based landmarks and random-projection landmarks lies in how they treat points in regions of heterogeneous density.

Clustering-based/sample-based landmarks (e.g., $k$-means centroids or representative data points) adapt to empirical density: dense regions receive many landmarks, while sparse regions receive few or none. Consequently, for a point $x$ located in a sparse region, the initial candidate set produced by clustering-based landmarks is typically dominated by points from nearby dense regions. This induces a *local optimality trap*: neighbor expansion primarily explores within dense clusters and fails to reach geometrically closer but sparsely supported neighbors. This explains the sample-based landmarks used in PyNNDescent has provided lower recalls than the random initialization in some datasets.

In contrast, random-projection landmarks are *density-oblivious*. Each landmark corresponds to an extreme direction rather than a region of high point mass. A point $x$ in a sparse region can still achieve an extreme projection value along some random direction, independently of local density. Conditioning on this event induces alignment of the projection with $x$, increasing the probability of co-occurrence with geometrically nearby points, regardless of their local density.

As a result, random-projection landmarks provide *uniform geometric coverage* across both dense and sparse regions. This avoids the dense–sparse discovery gap inherent to clustering-based landmarks and ensures that NNDescent does not become trapped in density-induced local optima, particularly for points lying in sparse or low-support regions of the space. The following lemma justifies this observation.

**Lemma 4.8** (Informal: Density Bias vs Directional Extremeness). *Let $X \subset \mathbb{S}^{d-1}$ be a dataset with heterogeneous density, and let $x \in X$ be a point lying in a sparse region, meaning that the local density of $X$ in a neighborhood of $x$ is asymptotically smaller than the global average.*

*Then:*

*(i) Under clustering-based landmark selection (e.g., $k$-means centroids or representative data points), the probability that a landmark lies within distance $O(d(x, y^*))$ of $x$ is negligible unless the number of landmarks scales with the inverse local density at $x$.*

*(ii) Under random-projection landmark selection, there exists a constant probability (independent of local density) that $x$ attains an extreme projection value along at least one landmark direction, thereby participating in a landmark bucket centered near $x$.*

*Consequently, clustering-based landmarks tend to underserve sparse regions, while random-projection landmarks provide uniform geometric coverage.*

### 4.4. From Theory to Practice

Our theoretical analysis requires a sufficiently large number of random vectors, i.e., $D = n/m$, as if $x$ is closest to $r_*$, i.e., $\langle x, r_* \rangle \approx 2\sqrt{\log D}$, then $x$ tends to be in Top-$m(r_*)$. However, this setting causes significant computation in random projection. To handle this hurdle, we use the "concatenation trick" used in locality-sensitive hashing implementations (Andoni et al., 2015; Pham & Liu, 2022).

Particularly, we consider two independent sets of random projection vectors $R = \{r_1, \ldots, r_{D'}\}, S = \{s_1, \ldots, s_{D'}\}$, where the coordinates of each vector $r_p, s_q$ are randomly selected from $N(0, 1)$. For each pair of random vectors $(r_p, s_q) \in [D'^2]$, we define a composite projection direction $w_{pq} = r_p + s_q$, and associate it with a bucket $B_{pq}$ storing the top-$m$ points $\mathbf{x} \in X$ with the largest score $x^\top w_{pq} = x^\top r_p + x^\top s_q$. This process can be executed efficiently by computing the top-$i$ closest vectors in $R$ and the top-$i$ closest vectors in $S$ to $x$, finding and hashing $x$ into the top-$i$ buckets.

Consequently, a setting of $D' = 512$ leads to $D \approx n/m$, which is sufficient to capture the geometric structure of million-point datasets. Furthermore, employing multiple landmark vectors per point $x$ acts as a *multi-probing* strategy,

significantly increasing the probability of $x$ being successfully hashed. This redundancy enhances the global connectivity and overall fidelity of the initial $c$-approximate $k$NN graph.

## 5. Experiment

We implement RPL in C++ and compile with `g++ -O3 -std=c++17 -fopenmp -march=native`. All experiments are conducted on an Ubuntu 20.04.4 server equipped with an AMD Ryzen Threadripper 3970X CPU (32 threads) and 128 GB RAM. Specifically, we design experiments to answer the following questions:

**Q1** Does RPL scheme improve the convergence of NNDescent over RPT and random initialization?

**Q2** How do key parameters (i.e., $k, m, i, D, L$) affect accuracy and computation cost?

We conduct experiments primarily under cosine similarity. We evaluate convergence behavior using two widely adopted metrics: (1) Recall@$k$ to measure graph accuracy. (2) The average number distance computations per point (both in initialization and NNDescent refinement). Following standard $k$NN graph construction benchmarks, we set $k = 50$, and report results averaged over 5 runs for stability. The results of other values of $k$ are provided in the appendix.

We perform experiments on four million-point datasets with high-dimensional deep embeddings:

- **GloVe-200** ($n = 1.18$M, $d = 200$) word embeddings from Common Crawl.

- **Word2Vec-300** ($n = 1.26$M, $d = 300$) Google News word embeddings.

- **CodeSearchNet-768** ($n = 1.37$M, $d = 768$) code representation embeddings.

- **Landmark-DINO-768** ($n = 0.76$M, $d = 768$) vision embedding dataset.

As RPL is complementary to existing NNDescent optimizations and libraries, we compare RPL against RPT and random initialization schemes with `PyNNDescent`, a widely used library for NNDescent in industry.

RPL and RPT share a similar mechanism that partitions data into several partitions (leaves of RPT and buckets of RPL) and runs a local join on each partition. We fix the size of these partitions for the sake of comparison, i.e., RPT uses `leafSize = m`. We also repeat $L$ times for RPT (trees) and RPL (tables). All methods share identical internal NNDescent refinement parameters.

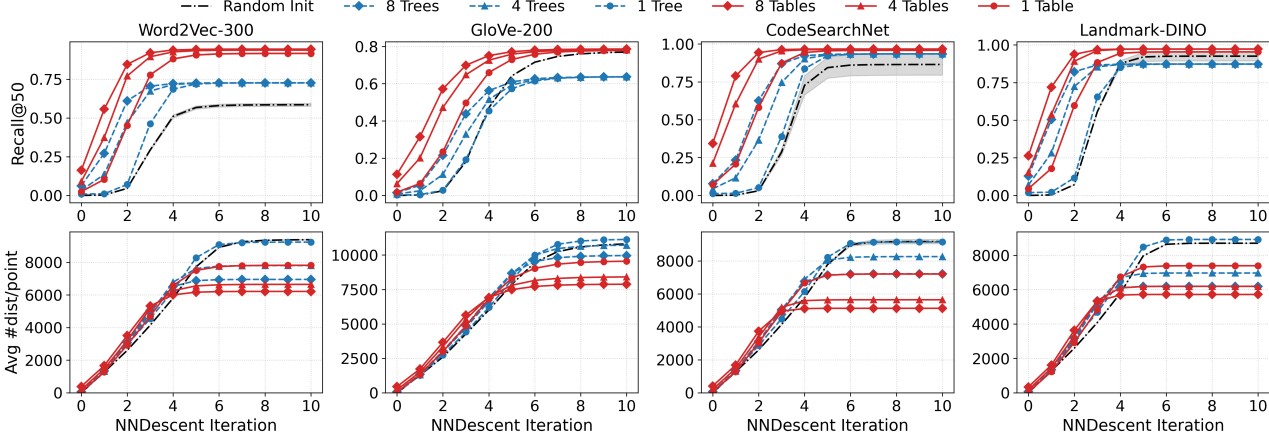

*Figure 1.* Recall@50 and average number of distance computation per point of NNDescent with RPL (Tables) and RPT (Trees), and random initialization schemes with $k = 50, m = 25$.

## 5.1. Main Results

Unless otherwise stated, we use $m = 25, i = 3, D' = 512$ for all experiments, and its effect is analyzed separately in the sensitivity study in Section 5.2.

**Comparison with RPT initialization.** Figures 1 shows the performance of NNDescent with different initialization strategies under the same setting for $k = 50, m = 25$. It reveals a consistent difference in both convergence behavior and computational cost between RPT and RPL, and random initializations.

The figure demonstrates that RPL constructs a superior initial graph for NNDescent compared to RPT. Specifically, the accuracy and number of distance computation of RPL scales more favorably with the number of tables $L$ across four datasets. This finding validates the structural advantages predicted by Theorem 4.2, confirming that RPL more effectively captures the metric space's underlying geometry.

Both RPL and RPT initializations improve recall rapidly within the first few NNDescent iterations, but RPT stalls early, resulting in noticeably nearly 20% lower final recall on Word2Vec and GloVe compared to RPL. RPT's recall ratios until convergence remain below RPL on CodeSearch-Net and Landmark-DINO.

In contrast, RPL reaches high recall within a few iterations and continues to refine effectively, achieving the highest final recall across all four datasets. NNDescent with RPL and RPT converges faster with larger $L$ though the difference is not significant between $L = 4$ and $L = 8$.

Even at $L = 8$, RPT exhibits significantly lower accuracy than random initialization on GloVe and Landmark-DINO, whereas RPL consistently achieves higher recall. This discrepancy exposes the local optimality trap inherent in data-dependent landmark schemes and underscores the structural

*Table 1.* Recall and runtime in second of NNDescent (RPT and RPL) and Faiss (IVF, IVFPQ) on Word2Vec with $L = 8, k = 50, m = 25$. We use 6 NNDescent iterations for RPL and RPT.

| Method | RPL | RPT | IVF | IVFPQ |
|---|---|---|---|---|
| Index | 12 | 6 | 27 | 36 |
| Query | 122 | 149 | 138 | 103 |
| Total | **134** | 155 | 165 | 139 |
| Recall@50 | **0.94** | 0.73 | 0.57 | 0.58 |

advantage of RPL's density-oblivious property in navigating complex data distributions.

RPL and RPT also have different convergence ratio, and hence different distance computation cost when running NNDescent. While RPL stabilizes both recall and average distance computations per point early, RPT continues to incur additional distance computations with little improvement in recall, particularly on Word2Vec and GloVe.

While the final recall of RPT eventually approaches that of RPL on datasets like CodeSearchNet, RPT incurs a substantially higher computational overhead due to a prolonged convergence phase. The efficiency of RPL in reaching near-optimal recall with fewer iterations confirms the merits of the initial $c$-approximate $k$NN graph and the rapid convergence properties established in our theoretical framework.

**Comparison with Faiss.** We carry out experiment to compare the performance of NNDescent with ANNS baselines, including Faiss-IVF and Faiss-IVFPQ, on constructing $k$NN graphs. We select Faiss as it balances the indexing and querying time while graph-based indexing methods have sufficient indexing time. We tune Faiss's parameters to achieve competitive recall under the time constraints, and detailed configurations are reported in the appendix.

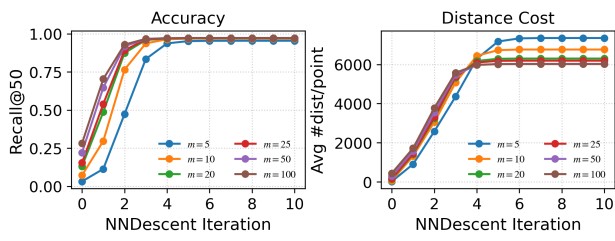

*Figure 2.* The sensitivity of $m$ of NNDescent with RPL on Landmark-DINO with $k = 50, i = 3, L = 4$.

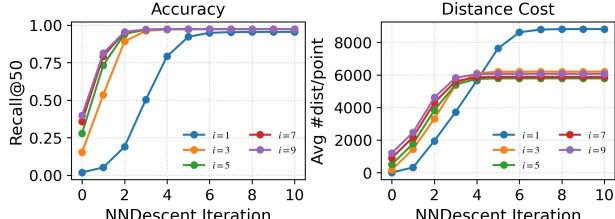

*Figure 3.* The sensitivity of $i$ of NNDescent with RPL on Landmark-DINO with $k = 50, m = 25, L = 4$.

For RPT and RPL, we report the initialization time as the index time and NNDescent refinement as query time. We present the results on Word2Vec as representative. Additional results on $k = 20$, and on the other dataset are provided in the appendix.

Table 1 shows RPL achieves the highest recall@50 of 0.94 while RPT is the second with 0.73 compared to under 0.60 provided by Faiss variants. Despite requiring larger indexing time due to the cost of constructing tables and executing local join on $D$ buckets, RPL's total running time still faster than RPT as it converges faster.

### 5.2. Sensitivity study

We carry out experiment to study the sensitivity of parameters $m$ and $i$ for RPL. We defer the study of $D, L$ to the appendix. Overall, the recall of NNDescent with RPL is not sensitive to these parameters. They slightly affect the number of distance computations, and hence the runtime of NNDescent.

**Sensitivity to $m$.** Figure 2 examines the effect of the neighborhood expansion parameter $m$ in RPL on Landmark-DINO with $k = 50$. As shown in the figure, increasing $m$ consistently improves recall, particularly in the early NNDescent iterations, but also leads to higher distance computation cost.

Larger values of $m$ accelerate convergence and achieve higher final recall. However, the improvement becomes marginal once $m \geq 25$, and the number of NNDescent iterations until convergence is the same. In contrast, $m < 25$ requires more distance throughout the refinement process due to a slower convergence though such difference is also maginal.

Overall, $m = 25$ offers a favorable balance between accuracy and distance cost, and we adopt this setting as the default in our main experiments.

**Sensitivity to $i$.** Different from RPT, RPL allows a point to be appear in several buckets, governing by the parameter $i$. Such multiple landmark assignment strategy will increase the accuracy of the initial graph with a negligible computational overhead.

Figure 3 shows that increasing $i$ significantly improves recall in the initial graph and very early NNDescent iterations. As we hash $i \cdot n$ points into a RPL tables, the bucket size tends to reach the limit of $m$ points, increasing the accuracy of the initial graph especially for the points in sparse areas. After a few iterations, the discrepancy is negligible when $i \geq 3$ as the NNDescent's outcome is already navigated by random vector landmarks aligned with the point itself.

We observe that the cost of constructing the initial graph with multiple landmark assignments is dominated by the NNDescent cost with $k = 50$. However, such cost is more significant with $k = 20$ as NNDescent's runtime is proportional to $k$. Therefore, we use a moderate number of multiple landmarks $i = 3$ to achieve a favorable balance between fast convergence, high recall, and distance efficiency for various $k$ values.

We observe that the performance of NNDescent with RPL is also stable with other parameters, including the number of random vectors $D$, and number of tables $L$. The experiment regarding these parameters is left in the appendix.

## 6. Conclusion and Future Work

In this paper, we demonstrated that NNDescent initialized with Random-Projection Landmarks (RPL) achieves a superior convergence rate of $O(d_0 \log c)$ by utilizing a "bridge" mechanism that guarantees geometric contraction toward the nearest neighbor. Our findings highlight that RPL's density-oblivious nature avoids the local optimality traps common in data-dependent schemes, ensuring robust global connectivity and reduced distance computation costs on large-scale datasets.

Moving forward, we aim to extend this framework to other metric spaces, e.g., $\ell_1$ and $\ell_2$, by employing kernel embedding techniques to map non-inner-product distances into high-dimensional Hilbert spaces where the RPL-bridge logic remains analytically valid.

## Impact Statement

This paper presents work whose goal is to advance the field of Machine Learning. There are many potential societal consequences of our work, none which we feel must be specifically highlighted here.

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

## A. Appendix: Missing proofs

### A.1. Missing proofs in Section 2.2

We define the used notation $o_p(\cdot)$ in the paper. Let $X_n$ be a sequence of random variables and $a_n$ be a sequence of constants. We say that:

$$X_n = o_p(a_n)$$

if for every $\epsilon > 0$:

$$\lim_{n \to \infty} P\left(\left|\frac{X_n}{a_n}\right| \geq \epsilon\right) = 0$$

In other words, the ratio $X_n/a_n$ converges in probability to zero.

**Lemma A.1** (Extreme Projection Decomposition). *Let* $\rho = \langle x, y \rangle \in [-1, 1]$. *Then, as* $D \to \infty$,

$$\langle y, r_* \rangle = \rho\sqrt{2 \log D} + \sqrt{1 - \rho^2}\, Z + o_p(\sqrt{\log D}),$$

*where* $Z \sim \mathcal{N}(0, 1)$.

A similar proof of this lemma can be found in (Pham, 2021; Xu & Pham, 2024) that use the concomitants of extreme order statistics.

*Proof.* For each $i$, decompose $r_i$ into components parallel and orthogonal to $x$:

$$r_i = \langle r_i, x \rangle x + r_i^\perp,$$

where $\langle r_i, x \rangle \sim \mathcal{N}(0, 1)$ and $r_i^\perp \sim \mathcal{N}(0, I_{d-1})$ is independent of $\langle r_i, x \rangle$.

Taking the inner product with $y$ yields

$$\langle y, r_i \rangle = \rho \langle x, r_i \rangle + \langle y_\perp, r_i^\perp \rangle,$$

where $y_\perp = y - \rho x$ satisfies $\|y_\perp\| = \sqrt{1 - \rho^2}$. Hence,

$$\langle y, r_i \rangle = \rho X_i + \sqrt{1 - \rho^2}\, Z_i,$$

with $X_i, Z_i \overset{\text{i.i.d.}}{\sim} \mathcal{N}(0, 1)$.

Let

$$M_D = \max_{1 \leq i \leq D} X_i = \langle x, r_* \rangle.$$

By classical extreme-value theory for Gaussian maxima,

$$M_D = \sqrt{2 \log D} - \frac{\log \log D + \log(4\pi)}{2\sqrt{2 \log D}} + o_p(1).$$

In particular,

$$M_D = \sqrt{2 \log D} + o_p(\sqrt{\log D}).$$

Substituting into the expression for $\langle y, r_* \rangle$ yields

$$\langle y, r_* \rangle = \rho\sqrt{2 \log D} + \sqrt{1 - \rho^2}\, Z + o_p(\sqrt{\log D}),$$

where $Z \sim \mathcal{N}(0, 1)$. □

**Lemma A.2** (Expectation of the $m$-th Largest Gaussian). *For any* $1 \leq m \leq n$,

$$\mathbb{E}[Z_{(m)}] \leq \sqrt{2 \log(n/m)} + O(1).$$

*Proof.* For $t \geq 0$, define the random variable

$$N(t) = \sum_{i=1}^{n} \mathbf{1}\{Z_i \geq t\},$$

the number of coordinates exceeding level $t$. Observe that

$$Z_{(m)} \geq t \quad \iff \quad N(t) \geq m.$$

By linearity of expectation and marginal Gaussianity,

$$\mathbb{E}[N(t)] = \sum_{i=1}^{n} \Pr(Z_i \geq t) = n \Pr(Z \geq t),$$

where $Z \sim \mathcal{N}(0, 1)$. Using the standard Gaussian tail bound,

$$\Pr(Z \geq t) \leq e^{-t^2/2},$$

we obtain

$$\mathbb{E}[N(t)] \leq n e^{-t^2/2}.$$

Let

$$t_m = \sqrt{2 \log(n/m)}.$$

Then $\mathbb{E}[N(t_m)] = m$.

Using Markov's inequality,

$$\Pr(Z_{(m)} \geq t) = \Pr(N(t) \geq m) \leq \frac{\mathbb{E}[N(t)]}{m}.$$

Therefore,

$$\mathbb{E}[Z_{(m)}] = \int_0^\infty \Pr(Z_{(m)} \geq t) \, dt \leq t_m + \int_{t_m}^\infty \frac{\mathbb{E}[N(t)]}{m} \, dt.$$

For $t \geq t_m$,

$$\frac{\mathbb{E}[N(t)]}{m} \leq \exp\left(-\frac{t^2 - t_m^2}{2}\right),$$

and the remaining integral is bounded by a constant independent of $n$. Hence,

$$\mathbb{E}[Z_{(m)}] \leq \sqrt{2 \log(n/m)} + O(1),$$

which completes the proof. $\square$

### A.2. Missing proofs in Section 4

**Lemma A.3** (Top-$m$ Inclusion Probability). *As $D \to \infty$, the probability that $y$ lies among the top-$m$ data points ranked by $\langle r_*, \cdot \rangle$ satisfies*

$$\Pr[y \in \text{Top-}m(r_*)] = \tag{1}$$

$$\Pr\left[Z \geq \frac{\sqrt{2 \log(n/m)} - \rho\sqrt{2 \log D}}{\sqrt{1 - \rho^2}}\right] + o(1), \tag{2}$$

*where $Z \sim \mathcal{N}(0, 1)$.*

*Proof sketch.* By extreme-value theory for Gaussian maxima,

$$\langle x, r_* \rangle = \sqrt{2 \log D} + o_p(\sqrt{\log D}).$$

For any fixed data point $y$ with $\langle x, y \rangle = \rho$,

$$\langle y, r_* \rangle = \rho \sqrt{2 \log D} + \sqrt{1 - \rho^2}\, Z + o_p(\sqrt{\log D}),$$

where $Z \sim \mathcal{N}(0, 1)$.

Let $t_m$ denote the random threshold corresponding to the $m$-th largest projection value among $\{\langle r_*, x_j \rangle\}_{j=1}^n$. By standard bounds on Gaussian order statistics, $t_m = \sqrt{2 \log(n/m)} + O_p(1)$. Substituting and rearranging yields the stated Gaussian tail probability. $\square$

**Theorem A.4** (Random-Projection Landmark Inclusion Probability). *Let $x \in \mathbb{S}^{d-1}$ be a fixed query point and let $y \in X = \{x_1, \ldots, x_n\} \subset \mathbb{S}^{d-1}$ be data points with $\rho = \langle x, y \rangle \in (0, 1)$. Let $r_1, \ldots, r_D \sim \mathcal{N}(0, I_d)$ be i.i.d. Gaussian random vectors, with $D = n/m$, and define*

$$r_* = \arg \max_{1 \le \ell \le D} \langle x, r_\ell \rangle.$$

*Let* $\text{Top-}m(r_*)$ *denote the $m$ points in $X$ with largest inner products with $r_*$.*

*Then, as $D \to \infty$, up to universal constants,*

$$\Pr[y \in \text{Top-}m(r_*)] = D^{-\frac{1-\rho}{1+\rho}}.$$

*Moreover, for any collection of points $\{y_i\}$ in $X$ with $\langle x, y_i \rangle \ge \rho$, the probability that at least one $y_i$ appears in $\text{Top-}m(r_*)$ is lower bounded by*

$$1 - \left(1 - D^{-\frac{1-\rho}{1+\rho}}\right)^{|\{y_i\}|},$$

*where dependence among projections can only increase this probability.*

*Proof.* It is standard that

$$\langle x, r_* \rangle = \sqrt{2 \log D} + O(1)$$

with high probability. Conditioned on $r_*$, joint Gaussianity implies

$$\langle y, r_* \rangle = \rho \langle x, r_* \rangle + \sqrt{1 - \rho^2}\, Z, \qquad Z \sim \mathcal{N}(0, 1).$$

The $m$-th largest projection among $n$ points satisfies

$$T_m = \sqrt{2 \log(n/m)} + O(1) = \sqrt{2 \log D} + O(1).$$

Hence $y \in \text{Top-}m(r_*)$ only if

$$\rho \sqrt{2 \log D} + \sqrt{1 - \rho^2}\, Z \ge \sqrt{2 \log D}.$$

Rearranging gives

$$Z \ge \sqrt{2 \log D} \sqrt{\frac{1 - \rho}{1 + \rho}}.$$

Applying the standard Gaussian tail bound when $t \to +\infty$,

$$\Pr[Z \ge t] = \exp\left(-\frac{t^2}{2}\left(1 \pm O(\tfrac{\log t}{t^2})\right)\right),$$

we have

$$\Pr[y \in \text{Top-}m(r_*)] \approx D^{-\frac{1-\rho}{1+\rho}}.$$

We now justify the extension to multiple points $\{y_i\}$ such that $\langle y_i, r \rangle \geq \rho$. W.l.o.g., we assume the set $\{y_i\}$ has size $s$. Conditioned on $r_*$, define the centered projections

$$W_i := \langle y_i, r_* \rangle - \rho \langle x, r_* \rangle, \qquad i = 1, \ldots, s.$$

By joint Gaussianity of $(r_*, x, y_1, \ldots, y_s)$, the vector $(W_1, \ldots, W_s)$ is jointly Gaussian with

$$\mathbb{E}[W_i] = 0, \qquad \mathrm{Var}(W_i) = 1 - \rho^2.$$

Moreover, for $i \neq j$,

$$\mathrm{Cov}(W_i, W_j) = \langle y_i, y_j \rangle - \rho_i \rho_j.$$

For points $y_i$ that are all near neighbors of $x$, we assume that $\langle y_i, y_j \rangle \geq \rho_i \rho_j$, implying $\mathrm{Cov}(W_i, W_j) \geq 0$. Hence $(W_1, \ldots, W_s)$ is a Gaussian vector with nonnegative pairwise correlations.

Consider the increasing event

$$\mathcal{E} = \left\{ \max_{1 \leq i \leq s} \langle y_i, r_* \rangle \geq T_m \right\} = \left\{ \max_i W_i \geq T_m - \rho \langle x, r_* \rangle \right\}.$$

Let $(\widetilde{W}_1, \ldots, \widetilde{W}_s)$ be independent Gaussian random variables with the same marginal distribution $\mathcal{N}(0, 1 - \rho^2)$. By Slepian's inequality, since

$$\mathrm{Cov}(W_i, W_j) \geq \mathrm{Cov}(\widetilde{W}_i, \widetilde{W}_j) = 0,$$

we obtain

$$\Pr(\mathcal{E}) \geq \Pr\left[ \max_i \widetilde{W}_i \geq T_m - \rho \langle x, r_* \rangle \right].$$

The right-hand side corresponds to the independent case and satisfies

$$\Pr\left[ \max_i \widetilde{W}_i \geq t \right] = 1 - \left( 1 - \Pr[\widetilde{W}_1 \geq t] \right)^s.$$

Substituting $t = \sqrt{2 \log D} \sqrt{\frac{1-\rho}{1+\rho}} + O(1)$ yields

$$\Pr(\mathcal{E}) \geq 1 - \left( 1 - D^{-\frac{1-\rho}{1+\rho}} \right)^s,$$

which proves the claim.

$\square$

**Theorem A.5** (Random-Projection Bridge Discovery). *Let $x \in \mathbb{S}^{d-1}$ be a query point and let $y \in X \subset \mathbb{S}^{d-1}$ be a discovered neighbor with $\langle x, y \rangle = \rho$. Define the improvement set*

$$\mathcal{Z}(x, y) = \{ z \in X : \langle x, z \rangle > \rho \}.$$

*Let $r \sim \mathcal{N}(0, I_d)$ be a random projection vector, and let $\mathrm{Top}\text{-}m(r)$ denote the set of points with the $m$ largest projection scores $\langle r, \cdot \rangle$.*

*Assume:*

*(A1) $y \in \mathrm{Top}\text{-}m(r)$.*

*(A2) There exists $\gamma > 0$ such that for all $z \in \mathcal{Z}(x, y)$,*

$$\langle y, z \rangle \geq \gamma.$$

*Then there exists a constant $c(\gamma) > 0$ such that*

$$\Pr[\exists z \in \mathcal{Z}(x, y) \text{ s.t. } z \in \mathrm{Top}\text{-}m(r) \mid y \in \mathrm{Top}\text{-}m(r)] \geq$$
$$1 - \exp\left( -c(\gamma) |\mathcal{Z}(x, y)| \right).$$

*In particular, the probability of discovering a strictly better neighbor increases monotonically with the number of improving neighbors.*

*Proof.* Condition on the event $\langle r, y \rangle = t$. By standard Gaussian conditioning,

$$r = ty + \xi, \qquad \xi \sim \mathcal{N}(0, I - yy^\top),$$

with $\xi$ independent of $t$.

For any $z \in \mathcal{Z}(x, y)$,

$$\langle r, z \rangle = t\langle y, z \rangle + \langle \xi, z \rangle,$$

where $\langle \xi, z \rangle \sim \mathcal{N}(0, 1 - \langle y, z \rangle^2)$.

By Assumption (A2),

$$\mathbb{E}[\langle r, z \rangle \mid \langle r, y \rangle = t] = t\langle y, z \rangle \ \geq \ \gamma t.$$

Let $\tau_m$ denote the (random) threshold for inclusion in Top-$m(r)$. Conditioning on $y \in$ Top-$m(r)$ implies $t \geq \tau_m$.

Hence for each $z \in \mathcal{Z}(x, y)$,

$$\Pr[\langle r, z \rangle \geq \tau_m \mid \langle r, y \rangle = t] \ \geq \ p(\gamma),$$

for some constant $p(\gamma) > 0$ depending only on $\gamma$ (using Gaussian tail bounds).

The collection $\{\langle r, z \rangle : z \in \mathcal{Z}(x, y)\}$ forms a positively correlated Gaussian vector (since all are increasing linear functionals of the same $r$). By Slepian's inequality, the probability that *none* of them exceeds $\tau_m$ is upper bounded by the corresponding independent case:

$$\Pr[\forall z \in \mathcal{Z}(x, y), \ \langle r, z \rangle < \tau_m \mid y \in \text{Top-}m(r)] \ \leq \ (1 - p(\gamma))^{|\mathcal{Z}(x, y)|}.$$

Therefore,

$$\Pr[\exists z \in \mathcal{Z}(x, y) \text{ in Top-}m(r) \mid y \in \text{Top-}m(r)] \ \geq$$
$$1 - \exp\big(-c(\gamma)|\mathcal{Z}(x, y)|\big),$$

for $c(\gamma) = -\log(1 - p(\gamma)) > 0$. $\qquad\square$

**Lemma A.6** (Informal: Density Bias vs Directional Extremeness). *Let $X \subset \mathbb{S}^{d-1}$ be a dataset with heterogeneous density, and let $x \in X$ be a point lying in a sparse region, meaning that the local density of $X$ in a neighborhood of $x$ is asymptotically smaller than the global average.*

*Then:*

*(i) Under clustering-based landmark selection (e.g., $k$-means centroids or representative data points), the probability that a landmark lies within distance $O(d(x, y^*))$ of $x$ is negligible unless the number of landmarks scales with the inverse local density at $x$.*

*(ii) Under random-projection landmark selection, there exists a constant probability (independent of local density) that $x$ attains an extreme projection value along at least one landmark direction, thereby participating in a landmark bucket centered near $x$.*

*Consequently, clustering-based landmarks tend to underserve sparse regions, while random-projection landmarks provide uniform geometric coverage.*

*Informal proof.* We argue each part separately.

**(i) Clustering-based landmarks.** Clustering objectives such as $k$-means minimize the expected squared distance to the data distribution. As a result, centroids concentrate in regions of high density. For a point $x$ in a sparse region, the expected distance to the nearest centroid is comparable to the distance to the nearest dense region, rather than to $d(x, y^*)$.

Unless the number of clusters grows inversely proportional to the local density at $x$, no centroid is likely to lie near $x$. Consequently, the candidate set generated for $x$ is dominated by points from dense regions, leading to exploration that is locally optimal with respect to density but suboptimal with respect to geometry.

**(ii) Random-projection landmarks.** Let $r \sim \mathcal{N}(0, I_d)$. The projection score $\langle r, x \rangle$ depends only on the direction of $x$, not on local point density. Across $D$ independent random projections, the maximum projection score of $x$ concentrates around $\sqrt{2 \log D}$, regardless of how many neighbors $x$ has.

Therefore, even if $x$ lies in a sparse region, with constant probability there exists a projection direction along which $x$ is among the top extreme points. Conditioning on this event induces a landmark bucket whose center aligns with $x$, enabling discovery of geometrically nearby neighbors independent of local density.

This establishes the claimed contrast between density-adaptive clustering and density-oblivious random projections. □

**A.3. NNDescent convergence for $k > 1$**

We extend Theorem 4.6 to the case $k > 1$.

**Theorem A.7** (High-Probability NNDescent Convergence). *Let $(X, d)$ be a metric space with doubling dimension $d_0$ and $|X| = n$. Initialize NNDescent using random-projection landmarks.*

*Then there exists a constant $C(d_0) > 0$ such that, with probability at least*

$$1 - n^{-C(d_0)},$$

*NNDescent converges to the exact kNN graph for all points in*

$$O(d_0 \log c_{\max})$$

*iterations, where $c_{\max}$ is the maximum initial approximation factor over all points.*

*Proof.* Fix a point $x \in X$. By Theorem 4.6, at each iteration where the current neighbor of $x$ is a $c$-approximation, the probability of discovering a strictly better neighbor is at least

$$p_c \geq 1 - \exp\left(-\eta(d_0) \log c\right).$$

Let $T_x = O(d_0 \log c_{\max})$ be the number of iterations required for convergence of point $x$. By independence of landmark projections across iterations and a union bound over the $T_x$ iterations,

$$\Pr[x \text{ fails to converge}] \leq \exp\left(-\Omega(\log n)\right).$$

Applying a union bound over all $n$ points,

$$\Pr[\exists x \in X \text{ that fails to converge}] \leq n \cdot \exp\left(-\Omega(\log n)\right)$$
$$= n^{-\Omega(1)}.$$

Thus, with probability at least $1 - n^{-C(d_0)}$, NNDescent converges for all points within the stated number of iterations. □

*Table 2.* Recall and runtime in second of NNDescent (RPT and RPL) and Faiss (IVF, IVFPQ) on CodeSearchNet with $L = 8, k = 50, m = 25$. We use 6 NNDescent iterations for RPL and RPT.

| Method | RPL | RPT | IVF | IVFPQ |
|--------|-----|-----|-----|-------|
| Index | 18 | 9 | 71 | 77 |
| Query | 186 | 227 | 690 | 148 |
| Total | **204** | 236 | 761 | 225 |
| Recall@50 | **0.97** | 0.93 | 0.82 | 0.69 |

# B. Appendix: Additional Experiments

### B.1. Parameter settings of Faiss

For all experiments, we use Faiss with $nlist = 8$. For search, we set $nprobe = 8$ for IVF and $nprobe = 32$ with $nbits = 8$ for IVFPQ. The number of PQ subquantizers is set to $m = 96$ for CodeSearchNet and $m = 100$ for Word2Vec, following common practice for high-dimensional embedding datasets.

### B.2. Other empirical results for $k = 50$

Table 2 reports the running time and recall@50 comparison on CodeSearchNet between NNDescent with RPL and RPT initialization and Faiss. RPL achieves the highest recall@50 of 0.97, outperforming RPT (0.93) and both Faiss variants. Although RPL requires additional indexing time due to table construction and local join operations, it converges faster during NNDescent refinement, resulting in a lower total running time than RPT.

Figure 4 shows the sensitivity of $m$ of RPT. The two RPT configurations ($m = 25$ and $m = 50$) deliver very similar performance, with only minor differences in both recall and distance computation across NNDescent iterations. In contrast, RPL with $m = 25$ consistently achieves higher recall with a lower distance cost, indicating a better recall–cost trade-off. Given the negligible gap between $m = 25$ and $m = 50$ for RPT, we use $m = 25$ as the default RPT setting in the remaining experiments.

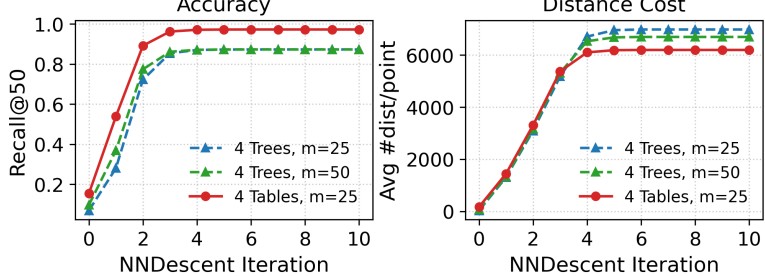

*Figure 4.* Sensitivity of $m$ of NNDescent with RPT on Landmark-DINO with $k = 50, L = 4$.

### B.3. Empirical results for $k = 20$

**Comparison with RPT scheme.** Figure 5 compares the recall@20 and distance computation cost of NNDescent under different initialization methods across four datasets. Random initialization converges slowly and consistently yields the lowest recall during early iterations. Both RPT and RPT significantly accelerate convergence, with RPL achieving the highest recall@20 across all datasets. Increasing the number of trees in RPT improves recall but also incurs higher distance cost, highlighting the inherent trade-off in tree based initialization. Moreover, RPL reaches comparable or better recall with substantially lower distance computation cost than RPT.

**Comparison with Faiss.** Table 3 and 4 show that RPL consistently achieves a more favorable accuracy and efficiency than both RPT and Faiss baselines. On Word2Vec, RPL improves recall@20 by 0.16 absolute over RPT (0.83 vs. 0.67) while requiring comparable total running time, and increases recall by nearly 48% relative compared to IVF and IVFPQ, with over 45% reduction in total runtime. On CodeSearchNet, RPL get a recall of 0.90, exceeding RPT by 4 points and reducing total

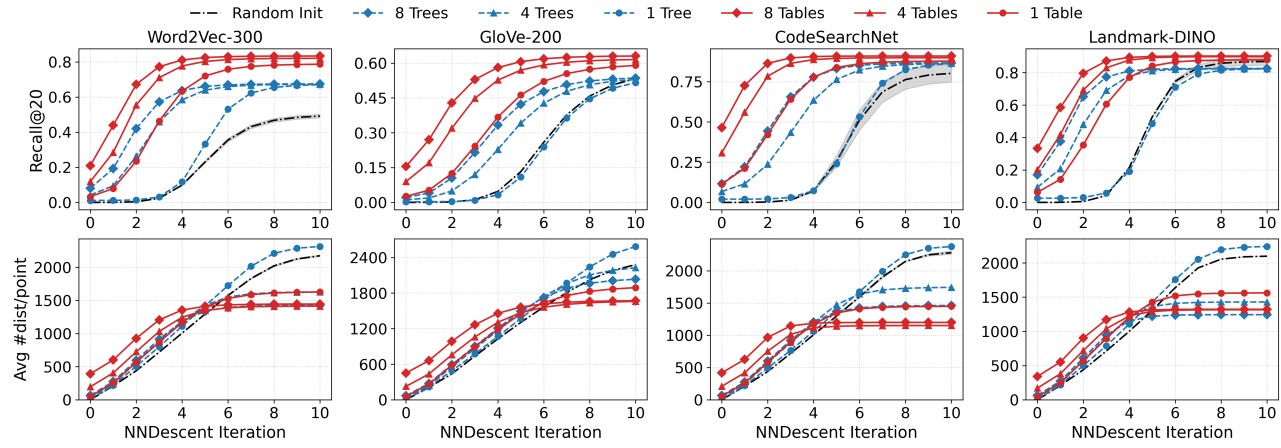

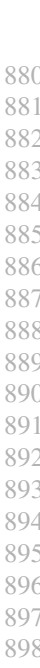

*Figure 5.* Recall@20 and average number of distance computation per point of NNDescent with RPL (Tables) and RPT (Trees), and random initialization schemes with $k = 20, m = 25$.

*Table 3.* Recall and runtime in second of NNDescent (RPT and RPL) and Faiss (IVF, IVFPQ) on Word2Vec with $L = 8, k = 20, m = 25$. We use 7 NNDescent iterations for RPL and RPT.

| Method | RPL | RPT | IVF | IVFPQ |
|---|---|---|---|---|
| Index | 13 | 6 | 27 | 36 |
| Query | 60 | 66 | 137 | 98 |
| Total | **73** | 72 | 164 | 134 |
| Recall@20 | **0.83** | 0.67 | 0.56 | 0.56 |

*Table 4.* Recall and runtime in second of NNDescent (RPT and RPL) and Faiss (IVF, IVFPQ) on CodeSearchNet with $L = 8, k = 20, m = 25$. We use 7 NNDescent iterations for RPL and RPT.

| Method | RPL | RPT | IVF | IVFPQ |
|---|---|---|---|---|
| Index | 18 | 9 | 69 | 77 |
| Query | 71 | 99 | 642 | 468 |
| Total | **89** | 108 | 711 | 545 |
| Recall@20 | **0.90** | 0.86 | 0.85 | 0.70 |

runtime by approximately 18% and delivers similar or higher recall than Faiss methods with over 80% lower total runtime.

## B.4. Sensitivity Study

**Sensitivity of $m$ with $k = 20$.** Figure 6 illustrates the sensitivity to the neighborhood size $m$ on Landmark-DINO. Compared to the case of $k = 50$, the refinement cost of NNDescent with $k = 20$ is substantially smaller, increasing $m$ leads to a more pronounced increase in the total distance computation, which directly affects the overall running time. However, $m = 25$ is still a relevant selection to balance the accuracy and runtime.

**Sensitivity of $i$ with $k = 20$.** Figure 7 shows that $i = 3$ provides a reasonable balance between accuracy and efficiency. While larger $i$ achieves similar recall after convergence, they incur consistently higher distance computation cost across iterations. In contrast, $i = 3$ achieves comparable final accuracy with the lowest overall cost, and we therefore adopt it as the default setting in the remaining experiments.

**Sensitivity of $L$ with $k = \{20, 50\}$.** Figure 8 and 9 show that increasing $L$ mainly accelerates early convergence for both $k = \{20, 50\}$ but also increases the distance computation, while the final recall remains largely unchanged. For $k = 20$, there is a significant increase in distance computation on constructing the intial graph by RPL, especially when $L > 8$

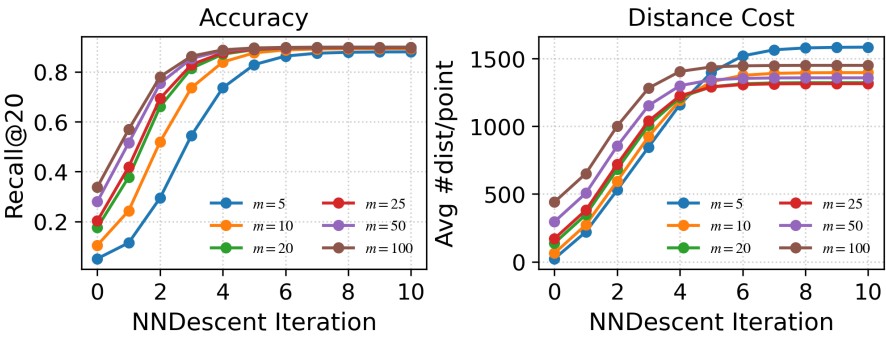

*Figure 6.* The sensitivity of $m$ of NNDescent with RPL on Landmark-DINO with $k = 20, i = 3, L = 4$.

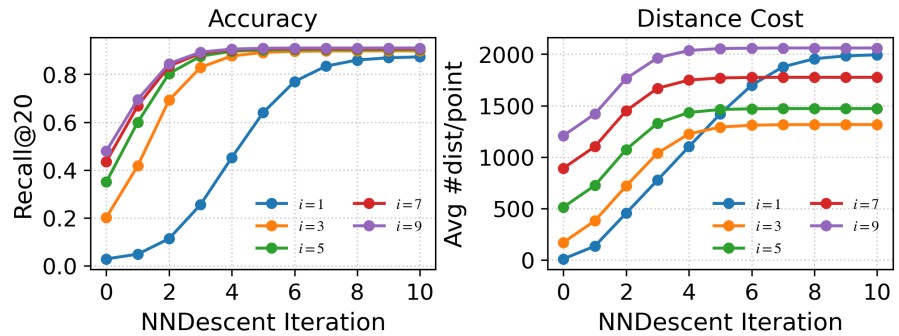

*Figure 7.* The sensitivity of $i$ of NNDescent with RPL on Landmark-DINO with $k = 20, m = 25, L = 4$.

though the final recall remains stable across different choices of $L$. Hence we select $L = 8$ in the main experiments.

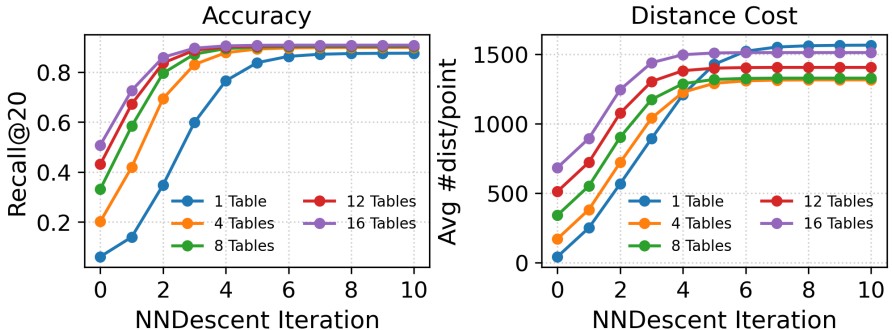

*Figure 8.* The sensitivity of $L$ of NNDescent with RPL on Landmark-DINO with $k = 20, i = 3, m = 25$.

**Sensitivity of $D$.** Figure 10 shows that different choices of $D'$ lead to very similar convergence behavior, final recall@20, and distance cost. Since increasing $D'$ beyond 512 does not provide noticeable improvement, we adopt $D' = 512$ as the default setting.

## C. A Heuristic Approach for Small $k$

We observe that for smaller $k$ (e.g., $k = 10$), the neighborhood size of "neighbor-of-neighbor" exploration is limited, affecting the accuracy of NNDescent. A simple solution is to run NNDescent to construct $k$NN graphs with larger $k$ (e.g., $k = 20$) and refine this graph to achieve higher quality graphs for smaller $k$.

Figure 11 shows that our heuristic significantly improves the recall@10 compared to the original NNDescent with $k = 10$ when using RPT and RPL. Regarding the distance computation, this heuristic increases slightly the runtime due to the

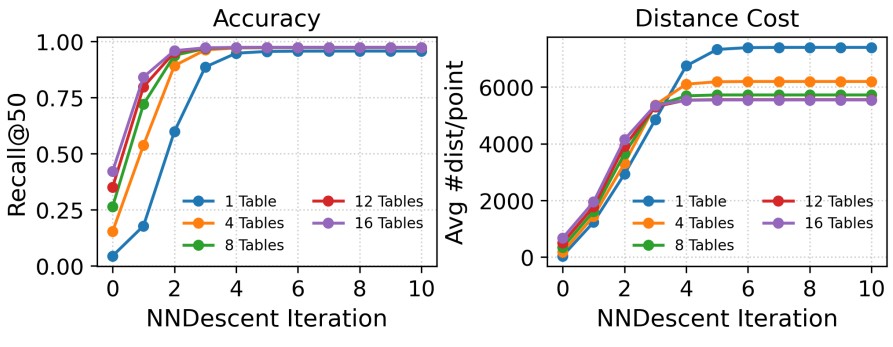

*Figure 9.* The sensitivity of $L$ of NNDescent with RPL on Landmark-DINO with $k = 50, i = 3, m = 25$.

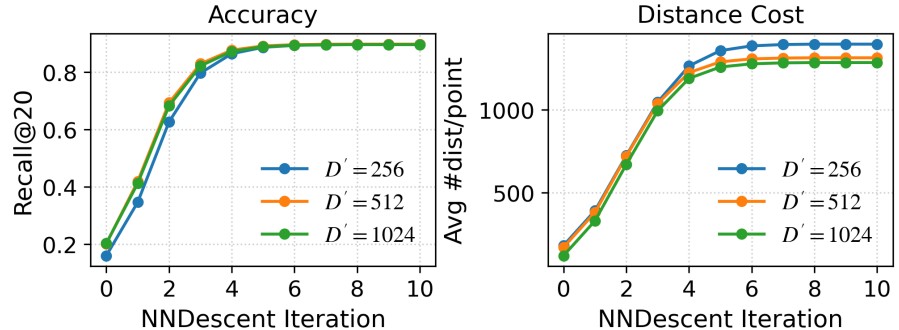

*Figure 10.* The sensitivity of $D$ of NNDescent with RPL on Landmark-DINO with $k = 20, i = 3, m = 25$.

increasing cost of internal local join operation of NNDescent.

## D. Comparison with LSH

We run a similar configurations of Falconn++ (Pham & Liu, 2022), an LSH representative methods that share similar mechanism with RPL as it can leverage the multi-probing strategy for indexing and querying. We also limit the LSH bucket size of $m$ points. Figure 12 shows that RLP achieves similar recall and convergence ratio as LSH while uses much less distance computation. This is due to the fact that Falconn++ has to evaluate distance computation for each point in all colliding buckets wheras RPL considers local join on each bucket, hence significantly reducing the number of distance computations.

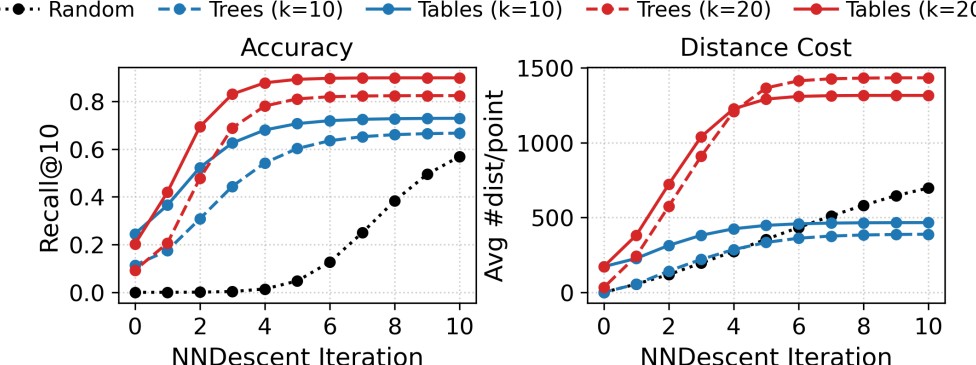

*Figure 11.* Recall@10 and average number of distance computation per point of NNDescent with RPL (Tables) and RPT (Trees), and random initialization schemes on Landmark-DINO with $i = 3, m = 25, L = 4$. We extract top-10 neighbor when $k = 20$ to compute recall@10.

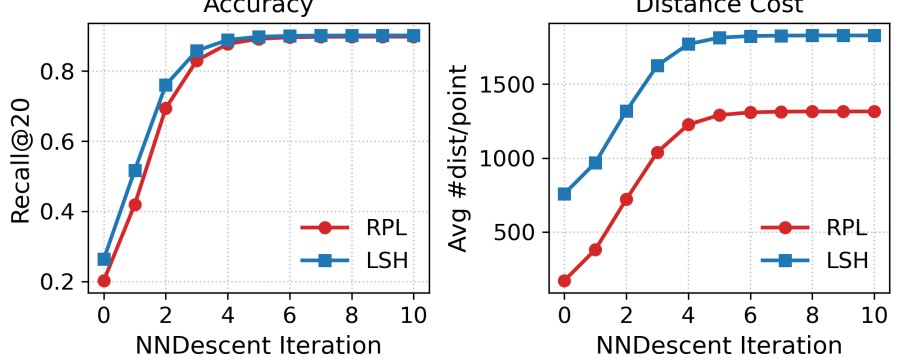

*Figure 12.* Comparing LSH (blue) and RPL over NNDescent iterations on Landmark-DINO with $L = 4, i = 3, m = 25$.

