# OpenReview forum: "Accelerating NNDescent with Random Projection Landmarks"
_ICML.cc/2026/Conference — Submitted to ICML 2026_

### Official Review · Reviewer_Uw2T · 2026-02-16

**Soundness:** 2
**Presentation:** 2
**Significance:** 2
**Originality:** 3
**Overall Recommendation:** 3
**Confidence:** 5

**Summary:**

This paper studies how to accelerate NNDescent by improving its initialization. The main idea is to use extreme random projections as “landmarks” (RPL). Specifically, instead of randomly initializing neighbors, the algorithm selects the Gaussian projection direction that aligns most strongly with the query point x and then uses that direction to generate candidate neighbors.

The authors give rich theoretical analyses to support this simple but effective intuition. The empirical results show the advantages of the proposed RPL over RPT and IVF-style baselines.

**Compliance With Llm Reviewing Policy:**

Affirmed.

**Key Questions For Authors:**

Please refer to the weakness part above

**Limitations:**

This paper has no formal discussion on their limitations. They have impact statement and I believe there is no negative social impact.

**Strengths And Weaknesses:**

# Strengths:

**(1) Simple but effective design:** The use of nearest random projections as directional proxies is conceptually clean and easy to understand. This idea, "such directions can bridge any points with their true neighbors", aligns with intuitions and sounds appealing.

**(2) Clear empirical improvements.** The experiments show a clear and consistent improvement of the proposed RPL over the chosen baselines.

**(3) Extensive theoretical analysis.** The provided analyses give useful insights on how NNdescent works in building k-nn graphs in an "iterative opimization" manner.


# Weaknesses:

However, I have a few concerns on the theoretical part and experiments, which may influence the foundation of this paper's results.

**Concern 1**: relying on a strong and unrealistic identity (assumption) $D=n/m$

The core inclusion probability in Theorem 4.2 is derived by aligning the two scales: $\langle x, r_* \rangle \approx \sqrt{2 \log D}$ and $Z_{(m)} \approx \sqrt{2 \log (n/m)}$. However, the practical implementation uses a fixed small D (e.g., 512) or small m (in contrast with any large n). This often violates the identity.

**Concern 2**: residual term distribution
Lemma 2.1 analyzes the selected landmark:
$$
r_*=\arg\max_{1\le \ell\le D}\langle x,r_\ell\rangle,
$$
so $r_*$ is a Gaussian vector **conditioned on an extreme event** and is NOT independent of $x$.
The proof has a decomposition of the form
$$
\langle y,r_*\rangle=\rho\langle x,r_*\rangle+\sqrt{1-\rho^2}Z,
\qquad \rho=\langle x,y\rangle,
$$
and treats the residual term has standard Gaussian behavior ($Z\sim N(0,\sigma)$) under this extreme selection. The paper does not explicitly justify the required conditional/joint distribution properties, which weakens the rigor of the subsequent probability calculations.


**Concern3**: Lemma 2.2 gives expectation but not concentration guarantee.

Lemma 2.2 derives

$$
\mathbb{E}[Z_{(m)}] \le \sqrt{2 \log (n/m)} + O(1),
$$

to be used in Theorem 4.2, but without assuming independence among the projections $Z_i$.

However, Theorem 4.2 requires that the Top-$m$ threshold satisfies

$$
Z_{(m)} \approx \sqrt{2 \log (n/m)}
\quad \text{with high probability}.
$$

Give the expectation of $Z_{(m)}$ is insufficient to guarantee concentration.
Expectation alone does not imply that the order statistic is sharply concentrated around its mean.
For example, $Z_i = \langle x_i, r_* \rangle$ may be dependent due to structure in the dataset (e.g., clustering or manifold alignment). Strong dependence can cause the order statistic $Z_{(m)}$ to deviate significantly from its nominal Gaussian threshold. This influence the probability calculation in Theorem 4.2, too.

**Concern 4**: influence of the $o(\sqrt{\log D})$ term

The proof uses the extreme-value estimate $ \langle x, r_* \rangle = \sqrt{2 \log D} + O(1)$, but the $O(1)$ term comes from the $o_p(\sqrt{log D})$ in Proof Lemma A.1. While this statement may be standard for Gaussian maxima, the subsequent derivation of Theorem 4.2 involves Gaussian tail calculations. In such calculations, even small perturbations of the threshold can affect the resulting power-law probability. In other words, if the deviation were only controlled as $o(\sqrt{\log D})$, rather than $O(1)$, then after the following derivation, the result may be completely different. This weakens the derived property of bridging x and y with landmark r.

*Concern 5**: the baseline choices

There are many methods in k-NN graph construction after NNdecent, such as [1-5]. This paper only chooses limited methods which are not state-of-the-art in their main experiments. Given there are plenty of GPU-based acceleration methods, it is hard to position the practical value of this paper in this context.

**Overall**, this paper seems neither strong enough from a theoretical perspective nor an empirical view.

references:

[1] Efanna: An extremely fast approximate nearest neighbor search algorithm based on knn graph

[2] FNNG: A High-Performance FPGA-based Accelerator for K-Nearest Neighbor Graph Construction

[3] Fast k-NN Graph Construction by GPU based NN-Descent

[4] Dynamic NN-Descent: An Efficient k-NN Graph Construction Method

[5] Efficient Distributed Approximate k-Nearest Neighbor Graph Construction by Multiway Random Division Forest

---

> ### Author Rebuttal · Authors · 2026-03-29
>
> We thank for your reviews and address your concerns below.
>
> __C1: Strong and unrealistic identity $D = n/m$ in Theorem 4.2 but the implementation use $D = 512$__
>
> Why our theoretical analysis based on $D = n/m$ random vectors, our practical implementation (see Sec 4.4) uses the tensor trick to simulate $D$ random vectors using 2 different sets $R$ and $S$, each has $D' = \sqrt{D} = \sqrt{n}$ random vectors. We consider the composite direction $w_{pq} = r_p + s_q$ where $r_p \in R, s_q \in S$ as a random vector in RPL. Hence, for each point $x$, we cast $r_* = (r_p, s_q)$ if $\( x^\top r_p + x^\top s_q \)$ is maximum.
>
> As the total number of pairs $(r_p, s_q)$ is $(D')^2$, setting $D' = 512$ (not $D = 512$) suffices for $D = n/m$ for $m \leq 50$ and $n$ is several million points.
>
> The cost of computing random projection boils down to $O(nd\sqrt{n})$ so not quadratic time any more. We use Fast Hadamard Transform (FHT) to simulate Gaussian random projections so the actual runtime of this step is negligible even compared to the the cost of join on each bucket of the table. The reason is FHT needs sequential memory accesses compared to expensive random memory accesses required by distance computation in joins.
>
> On Tab 1, the runtime of initializing kNN graph (index time) is less than 10\% of the total runtime.
>
> __C2: Proof of Lemma 2.1 (Appendix A1)__
>
> We do not need any required conditional/joint distribution properties to prove Lemma 2.1.
>
> We use the fact that for any standard Gaussian vector $r_i$ and _fixed_ unit vectors $x, y$, $(\langle x, r_i \rangle, \langle y, r_i \rangle)$ are bivariate variables from $N(0, 0, 1, 1, \rho)$, and hence $\langle y, r_i \rangle = \rho \langle x, r_i \rangle + \sqrt{1 - \rho^2} Z$ where $Z \sim N(0, 1)$ is independent of $\langle x, r_i \rangle$.
> Therefore, if $r_i$ is close to $x$, the behavior of the random variable $\langle x, r_i \rangle$ is more certain and so is $\langle y, r_i \rangle$. This is the standard result in extreme order statistics.
>
> **C3: Theorem 4.2 uses $Z_{(m)} \approx \sqrt{2 \log {(n/m)}}$**
>
> We are sorry to state the bound of $E[Z_{(m)}]$ but not using it.
> We only need the bound $Z_{(m)} \leq \sqrt{2 \log{n/m}} + O(1)$ to prove Theorem 4.2, and proving this bound is easier than the bound of $E[Z_{(m)}]$.
> On the proof of Lemma 2.2 in Appendix A.1 (L576). Using Markov inequality with $t = \sqrt{2\log(n/m)} + C$ for a constant $C > 0$, we can have $Z_{(m)} \leq \sqrt{2 \log{n/m}} + O(1)$ w.h.p.
>
> The threshold for inclusion in $\text{Top-}m(r_*)$ corresponds to the $m$-th largest projection among $n$ points, bounded by $t_m = \sqrt{2 \log(n/m)} + O(1) = \sqrt{2 \log D} + O(1)$.
> Thus, $y \in \text{Top-}m(r_*)$ occurs if:
> $$\rho \sqrt{2 \log D} + Z \sqrt{1-\rho^2} \ge t_m = \sqrt{2 \log D} + O(1) ,$$
> which yields the result of Theorem 4.2:
> $$p := \Pr[y \in \text{Top-}m(r_*)] \ge D^{-\frac{1-\rho}{1+\rho} + o(1)}.$$
>
> __C4: Influence of the $o_p(\sqrt{\log D})$ term__
>
> We agree that the current appendix does not justify the exact power-law exponent in Theorem 4.2 under only an $o_p(\log D)$ error term. Our intended use of Theorem 4.2 is not the exact exponent itself, but the existence of a nontrivial probability of retrieving a $c$-approximate neighbor, which can then be amplified through the candidate set $S = \\{ y_i \in X \mid \langle x, y_i \rangle = \rho_i \ge \rho \\}$.
>
> Clearly, larger $c$ leads to larger $|S|$. If we allow large $c$, the probability of finding such $y_i$ will be much larger and the influence of $o_p(\sqrt{\log D})$ term will be reduced. Indeed, the convergence of NNDescent is $O(\log c)$ in expectation so we think a larger $c$ from the initial kNN graph would not hurt much the convergence ratio of NNDescent in practice.
>
> __C5: The baseline choices__
>
> Our contribution is not a new end-to-end graph-construction system, but a better initialization strategy for NNDescent-style pipelines. For that reason, the experiments are designed to isolate the effect of initialization while holding the NNDescent refinement stage fixed, making PyNNDescent the most relevant primary baseline.
>
> NNDescent relies on local neighbor-of-neighbor refinement and can stagnate in __suboptimal__ graphs when initialization is poor (Fig. 1). Thus, while hardware acceleration improves runtime, it does not directly address the initialization bottleneck studied here. RPL does: it produces a stronger starting graph, which in turn yields both higher final accuracy and faster convergence. In this sense, our method is _complementary_ to hardware-accelerated NNDescent implementations.
>
> We agree that EFANNA is a relevant comparison point because it also uses tree-based candidate generation followed by NNDescent refinement, and we will clarify this relationship in the revision. PyNNDescent remains our primary baseline because it is a strong and widely used implementation in practice, including in popular UMAP and Scanpy libraries.

---

> > ### Author Rebuttal · Reviewer_Uw2T · 2026-04-03
> >
> > Thank you for the detailed rebuttal. It helps clarify several points.
> >
> > Your clarification about C1 is helpful, and it reduces my original concern to some extent. However, the theory is still stated for $D$ i.i.d. Gaussian directions, so the gap between the theorem and the actual implementation is reduced, but not fully closed.
> >
> > About C2, the issue is not the marginal bivariate Gaussian decomposition for a fixed \(r_i\), but whether the residual term remains valid after conditioning on the selected maximizer r*. I agree this may be fixable, but in the current form I do not think the rebuttal fully resolves the rigor concern.
> >
> > C3 can be seen as addressed.
> >
> > And I appreciate the honest clarification for C4, but this concern remains. In particular, your response seems to say that the current appendix does not justify the exact exponent in Theorem 4.2. That means the theorem still appears stronger than what is actually supported by the proof. This is still an important issue for me.
> >
> > For C5, I understand that this paper is mainly about initialization for NNDescent-style pipelines rather than a full end-to-end graph-construction pipeline. That said, I still think the experimental positioning is somewhat limited, and stronger or more relevant graph-construction baselines would help clarify the practical value more convincingly.
> >
> > Overall, I appreciate the clarifications, yet they are not strong enough for me to raise my score.

---

> > > ### Author Response · Authors · 2026-04-06
> > >
> > > We thank for your review and address your concerns below.
> > >
> > > __C2__, Let $(X_i = \langle x, r_i \rangle, Y_i = \langle y, r_i \rangle)$ be bivariate variables, we have $Y_i = \rho X_i + \sqrt{1 - \rho^2} Z_i$ where $X_i$ and $Z_i$ are independent and $Z_i \sim N(0, 1)$. Let $i_* = \arg \max X_i$. As $i_*$ is measurable with respect to $(X_1, \ldots, X_D)$, and $(Z_1, \ldots Z_D)$ are independent of $(X_1, \ldots, X_D)$, the selected residual $Z_{i_*}$ is still in N(0, 1) and independent of $X_{i_*}$. The result of Lemma 2.1 is the classic result in extreme order statistics, see David and Galambos, The asymptotic theory of concomitants of order statistics, Journal of Applied Probability, 1974.
> > >
> > > __C1__: In ML, there are many applications that require D i.i.d. Gaussian random vectors, and there is a long line of work on finding more efficient surrogates for these projections in practice. The key issue is exactly the gap between idealized Gaussian model vs. computational surrogate. From this perspective, insisting on implementing the exact theoretical construction can unnecessarily limit both innovation and practical impact. In many successful systems, the theoretical model serves as the conceptual foundation, while the deployed method uses a computationally efficient approximation.
> > > Representative examples include ANN/LSH systems (e.g. Andoni et al. 2015, Pham & Liu, 2022) which use tensorized or structured projection schemes as practical surrogates for Gaussian random projections. PyNNDescent is a widely used package in both CS and bioinformatics, even though its empirical success goes well beyond what is currently covered by formal analysis.
> > >
> > > __C4__:  To clarify the term $o_p(\sqrt{\log D})$ and other terms, we simplify the notation by defining $\epsilon_D = o_p(\sqrt{\log D})$. From Lemma 2.1: $\langle x, r_* \rangle = \sqrt{2 \log D} + \epsilon_D$, and $\langle y, r_* \rangle = \rho \sqrt{2 \log D} + \sqrt{1 - \rho^2} Z + \epsilon_D$ where $Z \sim N(0, 1)$.
> > >
> > > We keep the same proof structure as in the paper.
> > >
> > > Let $T_m = \sqrt{2 \log D} + O_p(1)$ with $D = n/m$, and define $t_D = \frac{T_m - \rho \sqrt{2 \log D} - \epsilon_D}{\sqrt{1 - \rho^2}}$. As $y \in \text{top-}m(r_*)$ if $\langle y, r_* \rangle \ge T_m$, we have $P[y \in \text{top-}m(r_*)] \ge P[Z \ge t_D]$.
> > >
> > > Define $\delta_D = \frac{T_m - \sqrt{2 \log D} - \epsilon_D}{\sqrt{1 - \rho^2}} = o_p(\sqrt{\log D})$ and $a = \sqrt{\frac{1-\rho}{1+\rho}}$, so $t_D = a \sqrt{2 \log D} + \delta_D$, and hence
> > > $$P[y \in \text{top-}m(r_*)] \ge P [ Z \ge t_D] =  P \left[ Z \ge a \sqrt{2 \log D} + \delta_D \right] .$$
> > >
> > > After simplifying this discovery probability, we will show its upper bound and lower bound.
> > >
> > > Since $\delta_D = o_p(\sqrt{\log D})$, there exists a deterministic sequence $\eta_D \to 0$ such that
> > > $P [ |\delta_D| \le \eta_D \sqrt{2 \log D} ] \to 1$. On the event $|\delta_D| \le \eta_D \sqrt{2 \log D}$, we have
> > > $(a - \eta_D) \sqrt{2\log D} \le a \sqrt{2\log D} + \delta_D \le (a + \eta_D) \sqrt{2\log D}$.
> > >
> > > Therefore, we have
> > >
> > > - Lower bound: $P [ y \in \text{top-}m(r_*)] \ge P[ Z \ge (a + \eta_D) \sqrt{2\log D} ] - o(1)$
> > >
> > > - Upper bound: $P [y \in \text{top-}m(r_*)] \le P [ Z \ge (a - \eta_D) \sqrt{2\log D} ] + o(1)$.
> > >
> > >
> > > Let $t_D^\pm = (a \pm \eta_D) \sqrt{2\log D}$, as $t_D^\pm \to \infty$, the Gaussian tail satisfies
> > > $P(Z \ge t_D^\pm) = \exp \left(-\frac{(t_D^\pm)^2}{2} + o \left((t_D^\pm)^2 \right) \right)$.
> > >
> > > Since $ (t_D^\pm)^2 =  \bigl(a^2 + o(1)\bigr) ( 2 \log D ) $ as $\eta_D \to 0$,
> > > $P [Z \ge t_D^\pm ] = \exp \left(-(a^2+o(1))\log D\right) = D^{-a^2+o(1)}.$
> > >
> > > Combining the upper and lower bounds yields
> > > $$P \left[y \in \text{top-}m(r_*) \right] = D^{-\frac{1-\rho}{1+\rho}+o(1)}.$$
> > >
> > > __C5__:  We agree that broader comparisons would strengthen the paper. However, the literature on high-quality kNN graph construction at million-point scale is relatively limited, and most practical pipelines are still centered around NNDescent-style refinement.
> > >
> > > - For [1] and [A], please see __W2__ in rebuttal to review 4ym4.
> > >
> > > - [4] improves the internal execution of NNDescent through better sampling in local-join operations.
> > >
> > > - [A, B] focuses on tree-based/LSH-based local graph construction without NNDescent refinement. Alternatively, they build index to answer n k-nearest neighbor queries. This line of works has generally been overtaken by stronger modern ANN libraries like Faiss. Our empirical results show that NNDescent pipelines provide higher quality kNN graph while sharing similar runtime to Faiss variants.
> > >
> > > - [C] is essentially the PyNNDescent algorithm.
> > >
> > > [1] Efanna: An extremely fast approximate nearest neighbor search algorithm based on knn graph
> > >
> > > [4] Dynamic NN-Descent: An Efficient k-NN Graph Construction Method
> > >
> > > [A] Wang et al. Scalable k-NN graph construction for visual descriptors, CVPR 2012
> > >
> > > [B] Zhang et al. Fast knn graph construction with locality sensitive hashing. ECML 13
> > >
> > > [C] Tange et al. Visualizing Large-scale and High-dimensional Data, WWW 16

---

### Official Review · Reviewer_ENLB · 2026-02-26

**Soundness:** 1
**Presentation:** 1
**Significance:** 2
**Originality:** 2
**Overall Recommendation:** 2
**Confidence:** 4

**Summary:**

The paper tries to improve NNDescent to build a k-nearest neighbor graph efficiently. It proposes Random Projection Landmarks (RPL), which mainly use the locality-sensitive hashing to initilize a rough neighbor graph and then optimize it using the orignial NNDescent. This paper spend lots of content attempting to prove that RPL can find $c$-approximate nearest neighbor with high probability within a bounded number of iterations. Empirical experiments on four million-scale datasets show that RPL requires fewer iterations and achieves faster convergence of NNDescent.

**Compliance With Llm Reviewing Policy:**

Affirmed.

**Final Justification:**

W3 > W2 > W1 > S1 > S2. The rebuttal only addresses part of the presentation issue. The more serious problems lie in the fact that the authors are unfamiliar with the related work and cannot properly illustrate and validate their contribution. Besides, both the paper and the rebuttal contain some potentially inaccurate content. Overall, considering that almost the whole paper needs to be reworked, I believe this manuscript is not ready, and I maintain my original score.

**Key Questions For Authors:**

See W1-W3

**Limitations:**

Don't see a explicit discussion of limitations in the paper. Main limitations can be seen in W1-W3

**Strengths And Weaknesses:**

**Strengths**

S1: kNN graphs do play an important role in information retrieval and machine learning. Algorithms that accelerate kNN graph construction may benefit a wide range of applications.

S2: Large-scale kNN graph construction commonly relies on NNDescent, which is a heuristic algorithm with limited theoretical guarantees. It is would be useful for practical system design and theoretical understanding when providing theoretical relation among the dataset size, approximation factor $c$, and the number of iterations

**Weaknesses**

W1: the presentation is poor. The narrative logic is relatively confusing and lacks coherence. Besides, some mathematical symbols are introduced without clear definitions or consistent usage. It is difficult to follow the algorithm design and theoretical analysis, such as

- Section 3: Which components are "analogous" to constructing $L$ hash tables and Which are not, and how does it affect the Algorithm 2? What does $n$ mean in the $O(dnDL+dm^2DL)$
- For worst case, is it possible that all $N$ points are in a same bucket and the complexity becomes $O(N^2)$?
- Algorithm 2: How is the the parameter $D$ used? What does Algorithm 2 line 4 exactly means?
- Section 4: how does the triangle inequality give out the equation in Line 218-219?
- There are inconsistencies such as: are $i,m,n$ used consistently throughout the paper? Line 320: are $x$ and $\mathbf{x}$ interchangeable? Besides, it is untraditional to use $i$ as a parameter of a alogirhtm.
- Line 308: should $D=n/m$ be $D=\lceil n/m \rceil$

W2: the experiment is unclear and insufficiently convincing.

- How is “querying” performed on the approximate kNN graph, and how many queries are conducted per dataset?
- Is the querying process identical across all algorithms (RPL, RPT, IVF, IVFPQ)?
- How many ground-truth nearest neighbors are computed per point, and what is the precise definition of Recall@$k$?
- The time cost is reported for only one dataset, and the improvement is marginal on the scale of second (e.g., RPL vs RPT: 134s vs 155s in Table 1). The efficiency gain appears small.
- The selection of baseline is kind of weird, considering that IVFPQ is for the quantization.

W3: the claimed contribution is not well justified and may be marginal..

- The paper contains potential factual inaccuracies. Such as
  - in Line 48: The Faiss is a library implementing various indexing algorithms, and kNN graph construction is not mandatory The algorithm HNSW does not require a kNN graph to build its index. ScaNN mainly focus on the quantization and also do not necessarily build a kNN graph.
  - The final iteration bound  $O(d_0\log c)$ is weird, since achieving a closer $c$-approximate nearest (smaller $c>1$) appears to require fewer iteration. Beside, the original NNDesent already converges very fast. It converges within just a few dozen times or even a few times in both original paper and the experiment in this paper. The proved bound might not be useful for practice.
- it lacks many related works, which weakens the claimed contribution. Random projection and its theoretical properties has been studied both in Euclidean distance and cosine similarity [1-4]. Besides pynndescent, using space partitioning or random projection to initialize a graph prior to NNDescent has also been explored in several works [5-7]. The paper does not clearly states how it differs from or improves upon these existing approaches.

[1] A. Andoni and P. Indyk, “Near-optimal hashing algorithms for approximate nearest neighbor in high dimensions,” *Commun. ACM*, vol. 51, no. 1, pp. 117–122, Jan. 2008, doi: [10.1145/1327452.1327494](https://doi.org/10.1145/1327452.1327494).

[2] M. Datar, N. Immorlica, P. Indyk, and V. S. Mirrokni, “Locality-sensitive hashing scheme based on p-stable distributions,” in *Proceedings of the twentieth annual symposium on Computational geometry - SCG ’04*, Brooklyn, New York, USA: ACM Press, 2004, p. 253. doi: [10.1145/997817.997857](https://doi.org/10.1145/997817.997857).

[3] M. S. Charikar, “Similarity estimation techniques from rounding algorithms,” in *Proceedings of the thiry-fourth annual ACM symposium on Theory of computing*, in STOC ’02. New York, NY, USA: Association for Computing Machinery, May 2002, pp. 380–388. doi: [10.1145/509907.509965](https://doi.org/10.1145/509907.509965).

[4] M. X. Goemans and D. P. Williamson, “Improved approximation algorithms for maximum cut and satisfiability problems using semidefinite programming,” *J. ACM*, vol. 42, no. 6, pp. 1115–1145, Nov. 1995, doi: [10.1145/227683.227684](https://doi.org/10.1145/227683.227684).

[5] R. Qiu and J. Tang, “Efficient Approximate Nearest Neighbor Search via Hemi-Sphere Centroids Graph,” *Proc. ACM Manag. Data*, vol. 3, no. 6, p. 321:1-321:26, Dec. 2025, doi: [10.1145/3769786](https://doi.org/10.1145/3769786).

[6] J. Wang, J. Wang, G. Zeng, Z. Tu, R. Gan, and S. Li, “Scalable k-NN graph construction for visual descriptors,” in *2012 IEEE Conference on Computer Vision and Pattern Recognition*, Providence, RI: IEEE, Jun. 2012, pp. 1106–1113. doi: [10.1109/CVPR.2012.6247790](https://doi.org/10.1109/CVPR.2012.6247790).

[7] C. Fu and D. Cai, “EFANNA : An Extremely Fast Approximate Nearest Neighbor Search Algorithm Based on kNN Graph,” Dec. 03, 2016, *arXiv*: arXiv:1609.07228. doi: [10.48550/arXiv.1609.07228](https://doi.org/10.48550/arXiv.1609.07228).

---

> ### Author Rebuttal · Authors · 2026-03-29
>
> We thank for your review and address your concerns below.
>
> **W1: Clarification of Algorithm 2**
>
> The whole Alg 2 can be seen as constructing 1 hash table and executing the kNN search for __all__ points on this table only. L3-5 is to compute $i$ buckets (represented by top-$i$ landmark random vectors) for each point $x$ and insert $x$ into these $i$ buckets.
> L7-14 show the search process. First, we prune each bucket associated to a random vector $r$ so that __each__ bucket has $m$ points, which are the top-m points closest to $r$. Then we run a join operation on each bucket to find kNN candidate for each point in the bucket. Since we run a join operation on each bucket, if $x$ appears in several buckets, it has better chance to refine its kNN candidates. If $x$ does not appear on any bucket (due to the bucket pruning procedure), its kNN is empty. However, by using $i > 1$ and repeating Alg 2 $L$ times, such case never happens in practice.
>
> $n$ is the size of the data set. For worst case, it is possible that $n$ points are in 1 bucket. But the complexity is not $O(n^2)$ as the bucket is pruned to have at most $m$ points. Hence, the complexity is $O(m^2)$ and majority of points will have empty kNN.
>
> An alternative view of L4 is to assign each point $x$ into $i$ closest cluster centers where the cluster center is represented by the a landmark random vector. We build inverted indexes where a random vector is used as the cluster center, each $x$ is assigned to top-$i$ closest cluster centers, and each cluster only keeps top-$m$ points closest to its center.
>
> __W2: Unclear experiment__
>
> On using Faiss (IVF, IVFPQ) to construct kNN graph, we have $n$ queries from the data set. We do not consider out-of-sample querying performance on the constructed kNN graph. We run bruteforce to compute exact kNN graph. The recall@$k$ is the average accuracy between approximate kNN graph and the exact kNN graph.
>
> On NNDescent variants, the query time is the NNDescent time. The index time is the initial kNN graph constructions with RPT and RPT (L397, left). On Tab 1, we try to find the configuration so that the runtime of all methods are similar to demonstrate the accuracy of NNDescent methods on constructing kNN graph. This is because downstream applications require high-quality kNN graph. From Tab 1, RPL outputs 94\% accuracy, more than 20\% higher than the other methods with similar runtime.
>
> We consider IVFPQ as its query time with $n$ queries is smaller than IVF. This is a trade-off between speed-accuracy in ANNS.
>
> __W3: The theoretical claims__
>
> Regarding the triangle inequality (L218): We realize a new technique to achieve similar lower bound for Theorem 4.2 without using this assumption. Details is on W3 rebuttal for the reviewer k3mH.
>
> Our main results show that __only__ the initialization phase with RPL (Alg 2) can achieve a $c$-approximate nearest neighbor for each point with some probality. But RPL cannot guarantee to have a small $c$ for each point. When using the output of Alg 2 with NNDescent, the number of NNDescent iterations will be $O(d_0 \log{c})$ in expectation, so a coarse kNN graph with large $c$ would not affect the convergence much. Of course, if RPL outputs good initial kNN graphs with small $c$, then NNDescent converges faster.
>
> We agree the original NNDescent converges very fast, but __not__ to the global optimum (Fig 1: Word2Vec, CodeSearch). We aim to characterize the convergence of NNDescent, and clarify assumptions that NNDescent with RPL converges to the global optimum.
>
> __W4. Related works and PyNNDescent baseline__
>
> We agree that random projections are widely use in ANNS [1-4]. However, none of them are designed to accelerate and characterize NNDescent convergence.
>
> - [5] indeed builds a graph index to support downstream ANNS. It cannot be used to build kNN graph as building graph index takes signifcantly time.
>
> - EFANNA [7] builds several randomized KD trees. It then executes a tree-guided candidate collection to initialize kNN for each point $x$, and hence form a coarse kNN graph. It then uses NNDescent to improve this graph in the second phase. EFANNA is indeed a variant of PyNNDescent where PyNNDescent uses several random projection trees to form a coarse kNN graph (via a join operation on each leaf), then runs NNDescent later on this coarse kNN.
>
> - [6] builds several random projection trees and run a join operation on each leaf to approximate the kNN graph. It is essential the first phase of PyNNDescent-RPT. It does not use NNDescent to refine the kNN graph so it outputs lower quality kNN graph than EFANNA.
>
> We compare to PyNNDescent, a strong baseline in industry. PyNNDescent is widely used in many applications/libraries, ranging from single-cell analysis ([A]), visualization (UMAP ecosystem [B]).
>
> - [A] SCANPY: large-scale single-cell gene expression data analysis. Genome Biol. 2018
>
> - [B] UMAP: Uniform Manifold Approximation and Projection for Dimension Reduction, 2018

---

> > ### Author Rebuttal · Reviewer_ENLB · 2026-04-02
> >
> > Thanks for the response and clarification. While the presentation could be improved in a revision,  I believe that other more serious issues cannot be adequately addressed within review cycle. The lack of engagement with related work, the unclear (and in my view, marginal) contributions, coupled with potentially inaccurate illustration and inconvincing experiment. These lead me to conclude that this manuscript is not ready and requires more substantial reworking.

---

> > > ### Author Response · Authors · 2026-04-05
> > >
> > > We thank for your comment. However, we respectfully note that our rebuttal addressed each of the concerns raised by the reviewer.
> > >
> > > __Lack of engagement with related work__
> > >
> > > See __W4__ of our rebuttal. There, we explained why ref [5] is not directly relevant to our setting, clarified that [7] is in fact closely related in spirit to our chosen baseline PyNNDescent, and explained why [6] corresponds only to an earlier tree-based stage and therefore yields lower graph quality than [7] and PyNNDescent. Thus, the rebuttal did engage directly with the cited related work and clarified our positioning with respect to it.
> > >
> > > __Unclear contribution__
> > >
> > > The reviewer’s concern here appears to stem from a misunderstanding of our theoretical and practical contributions, particularly the statement "The final iteration bound $O(d_0 \log c)$ is weird, since achieving a closer $c$-approximate nearest (smaller $c > 1$) appears to require fewer iteration".
> > >
> > > As we clarified in __W3__, the role of achieving a $c$-approximate nearest neighbor is only to obtain a stronger initial kNN graph through RPL. A smaller $c$ means a better initialization, which in turn leads to faster NNDescent convergence toward the global optimum. This is exactly consistent with our theory and intuition.
> > >
> > > As an evidence: the recall@50 on Fig 1 of all methods at __0 NNDescent iterations__ measures the accuracy of the initial kNN graph with different initialization algorithms. Under the same configurations, RPL consistently returns a better quality initial kNN graph than RPT and random initialization, already recovering nearly 20% of the exact 50 nearest neighbors before NNDescent refinement. Since NNDescent can converge to suboptimal graphs when initialization is weak, this stronger starting graph is practically important for downstream applications that require high-quality kNN graph construction.
> > >
> > > __Potentially inaccurate illustration__
> > >
> > > In __W1__, we clarified how RPL can be understood both in the language of LSH and in the language of Faiss-style indexing. The reviewer’s question "For worst case, is it possible that all $N$ points are in a same bucket and the complexity becomes $O(N^2)$?" suggests a misunderstanding of the key design of RPL: each bucket is explicitly pruned to contain at most $m$ points. This bounded bucket size is precisely what makes RPL efficient in practice, even though the resulting procedure is more subtle to analyze theoretically.
> > >
> > >
> > > __Inconvincing experiment__
> > >
> > > Our experiments are specifically designed to evaluate the competitiveness of NNDescent-RPL against NNDescent-RPT and standard NNDescent for the task of kNN graph construction. We measure graph quality using recall@$k$, and computational efficiency using the number of distance computations, which is the main determinant of runtime in these methods (see Lines 296–300, right). We do not study downstream applications such as similarity search over a graph index, which is instead the focus of the reviewer’s reference [5]. In that sense, the questions raised in __W2__ appear to reflect a misunderstanding of the experimental setting and evaluation protocol, both of which were clarified in detail in our rebuttal.
> > >
> > > __Overall__, we think the remaining disagreement appears to be less about unanswered rebuttal points and more about the reviewer’s overall judgment of the paper.

---

### Official Review · Reviewer_k3mH · 2026-03-09

**Soundness:** 1
**Presentation:** 2
**Significance:** 3
**Originality:** 2
**Overall Recommendation:** 2
**Confidence:** 4

**Summary:**

The submission provides a novel version of the NNDescent algorithm for approximate $k$-nearest neighbors search under cosine distances. The paper proposes to use so-called landmarks to obtain a better initialization for the NNDescent algorithm, which enables faster convergence to exact solutions. It also presents theoretical results for this algorithm.

**Compliance With Llm Reviewing Policy:**

Affirmed.

**Final Justification:**

After considering the rebuttal and reading the other reviews, I stick with my original score.

**Key Questions For Authors:**

1. Did you verify the assumption of Theorem 4.2 on real-world datasets?
1. How did you obtain the choice of $D'$?

**Limitations:**

Yes.

**Strengths And Weaknesses:**

**Strengths:**
- Approximating nearest neighbor graphs efficiently is highly important and a well-researched area. Therefore, positive results in this direction are highly relevant.
- The proposed algorithm contains interesting ideas.

**Weaknesses:**
- The statement of the theoretical results and the proofs is vague. Concrete examples for this are as follows:
    - Results like Lemma 4.1 are only proven for $D \to \infty$. However, this is quite undesirable since the algorithm's running time depends quadratically on $D$ (it is the number of landmarks). The dependency of $D$ on the number of points must be made explicit.
    - Theorem 4.2 claims that $P[y \in \text{Top-}m(r_*)] = D^{\frac{1-\rho}{1+\rho}}$. However, the theorem's proof only shows that $P[y \in \text{Top-}m(r_*)]  \approx D^{\frac{1-\rho}{1+\rho}}$ (see Line 658) and it is not quite clear to me how "$\approx$" should be interpreted formally.
    - The version of Theorem 4.2 stated in the appendix (Theorem A.4, Line 618) is different from the one in the main text. Particularly, the assumption on the dot products is missing.
- Theorem 4.2 relies on an assumption on the dot products of certain points. While some justification is given after the statement of Theorem 4.2, I do not find it very convincing.
- In several places, the paper's writeup lacks precision. Specifically:
     - The pseudocode for NNDescent in Algorithm 1 is missing important pruning steps.
     - The running time analysis in Line 106 is not correct since the size of the set $B(x)$ cannot be bounded by $O(k)$, since during the local join steps of NNDescent the size of $B(x)$ could get much bigger.
     - I cannot follow why setting $D' = 512$ leads to $D \approx n/m$ in Line 325. How was this particular choice of $D'$ obtained?

---

> ### Author Rebuttal · Authors · 2026-03-29
>
> We thank for your review and address your concerns below.
>
> __W1. The condition $D \rightarrow \infty$__
>
> This condition ensures $\langle x, r_*\rangle$ is dominated by the term $\sqrt{2 \log D}$. In the proof (L535): $\langle x, r_*\rangle = \sqrt{2 \log D} - \log{\log D} / 2\sqrt{2 \log D} + o_p(1) = \sqrt{2 \log D} + o_p(\sqrt{\log D})$. This standard result was widely used in many theoretical papers, e.g.
>
> - Lossy Compression via Sparse Linear Regression: Computationally Efficient Encoding and Decoding. IEEE Trans. Inf. Theory (2014)
>
> - Distributed Estimation of Gaussian Correlations. IEEE Trans. Inf. Theory (2019)
>
>
> We explicitly set $D = n/m$ in Theorem 4.2 (L196), which means that the term $o_p(\sqrt{\log D})$ will be very small when $n$ is large. Given $D = O(n)$ for a small $m$, Sec 4.4 shows how to break quadratic cost of random projections in practice (see details in W5).
>
> **W2. Claim $P[y \in \text{Top-}m(r_*)] \approx D^{-\frac{1-\rho}{1 + \rho}}$ in Theorem 4.2**
>
> On L642 in the proof of Theorem 4.2, the threshold for inclusion in $\text{Top-}m(r_*)$ corresponds to the $m$-th largest projection among $n$ points: $T_m = \sqrt{2 \log(n/m)} + O(1) = \sqrt{2 \log D} + O(1)$.
> Thus, $y \in \text{Top-}m(r_*)$ occurs if $\rho \sqrt{2 \log D} + Z\sqrt{1-\rho^2} \ge T_m = \sqrt{2 \log D} + O(1).$
> Then, $Z \ge \sqrt{2 \log D} \Big( \sqrt{\frac{1-\rho}{1+\rho}} +o(1) \Big)$. Applying the standard Gaussian tail bound, we obtain the discovery probability $p$:
> $$p := \Pr[y \in \text{Top-}m(r_*)] \ge D^{-\frac{1-\rho}{1+\rho} + o(1)}.$$
>
> __W3. Theorem 4.2 assumptions__
>
> After the submission, we realized that we do not need this assumption to lower bound the the probability of finding one of $y_i$ in the set $S = \\{ y_i \\} = \\{ y_i \in X \mid \langle x, y_i \rangle = \rho_i \ge \rho \\}$ __dependent__ on $|S|$. We use the second moment technique to achieve the following result.
>
> Let $p = D^{-\frac{1-\rho}{1+\rho} + o(1)}$ be the baseline discovery probability for similarity $\rho$, let $p_1 \ge p$ be the average _individual_ discovery probability in $S$, and let $p_2$ be the average _joint_ discovery probability for any _pair_ in $S$.
> The probability of at least one $y_i \in S$ appears in $\text{Top-}m(r_*)$ is strictly lower-bounded by:
> $P[\exists y_i \in S \text{ and }  y_i \in \text{Top-}m(r_*)] \ge \frac{p_1^2}{p_2 + (p_1 - p_2) / |S|}$.
>
> We sketch the proof here.
>
> Given indicator variable $X_i$ for each neighbor $y_i$, where $X_i = 1$ if $y_i \in \text{Top-}m(r_*)$ and $X_i = 0$ otherwise, we have $X = \sum_{i=1}^{|S|} X_i$ be the total number of neighbors from $S$ discovered. Since $X$ is a positive integer, by the Paley-Zygmund inequality, we lower-bound the probability of discovering at least one $y_i \in S$: $P[X \ge 1] \ge \frac{(E[X])^2}{E[X^2]}$. Then by computing $E[X]$ on $p_1$ and $E[X^2]$ on $p_2$, we can achieve the lower bound stated above.
> Importantly, this lower bound still depends on the size of $|S|$, so leading to good probability to discover $y_i$ if large $S$.
>
> __W4. Representation__
>
> _The pseudocode NNDescent missing pruning steps:_ As we aim to accelerate NNDescent with a better initial kNN graphs and use any NNDescent implementation as a blackbox, we decide to not elaborate NNDescent in details for space saving.
>
> _The running time of local join of NNDescent_: As $B(x)$ contains approximate kNN of $x$, so we can alway ensure $|B(x)| = O(k)$ after every NNDescent iteration.
>
> __W5. The setting of $D' = 512$ and $D$ in practice__
>
> Why our theoretical analysis bases on $D = n/m$ random vectors, our practical implementation (Sec 4.4) uses the tensor trick to simulate $D$ random vectors using 2 different sets $R$ and $S$, each has $D' = \sqrt{D} = \sqrt{n}$ random vectors. We consider the composite direction $w_{pq} = r_p + s_q$ where $r_p \in R, s_q \in S$ as a random vector in RPL.
>
> As the total number of pairs $(r_p, s_q)$ is $(D')^2$, $D' = 512$ suffices for $D = n/m$ for $m \leq 50$ and $n$ is several million points.
>
> The cost of computing random projection boils down to $O(nd\sqrt{n})$ so not quadratic time any more. We use Fast Hadamard Transform (FHT) to simulate Gaussian random projections so the actual runtime of this step is negligible even compared to the the cost of join on each bucket of the table. The reason is FHT needs sequential memory accesses compared to expensive random memory accesses required by distance computation in joins.
>
> On Tab 1, the runtime of initializing kNN graph (index time) is less than 10\% of the total runtime.
>
> __W6. Reliability of Theorem 4.2 on real-world datasets?__
>
> The recall@50 on Fig 1 of _all_ methods at __0__ NNDescent iteration measures the accuracy of the initial kNN graph with different algorithms. On the same configurations, RPL often returns a better quality initial kNN graph than RPT and random initialization. This graph indicates RPL already found nearly 20\% of the _exact_ 50 nearest neighbors on the initial kNN graph.

---

> > ### Author Rebuttal · Reviewer_k3mH · 2026-04-01
> >
> > Thank you for your detailed response.
> >
> > I appreciate your effort and am happy that you could remove the assumption in Theorem 4.2. However, I believe that such changes are more of a major revision and go beyond a simple rebuttal.
> >
> > Regarding W1, I would appreciate if you could rephrase the theorem statement to be more precise. Specifically, replace the $D \to \infty$ with some concrete statement that "if $D\geq f(n,m)$ (for some function $f(n,m)$ that you specify) then your claim holds".

---

> > > ### Author Response · Authors · 2026-04-05
> > >
> > > We thank you for your additional comments. Indeed, the lemma is correct for $D \rightarrow \infty$ due to the term $o_p(\sqrt{\log D})$. We will clarify that we set $D = f(n, m)$ to ensure $o_p(\sqrt{\log D})$ very small in practice.
> > >
> > > Based on your rebuttal acknowledgement, our understanding is that the rebuttal addresses essentially all of your raised concerns, except for __W3__ regarding the proof of Theorem 4.2 under the assumption you found unconvincing.
> > >
> > > As clarified in the rebuttal, this assumption was used only for the second part of Theorem 4.2, namely the lower bound on the probability of discovering a $c$-approximate nearest neighbor of a point $x$ as a function of the size of of the set
> > >
> > > $$S = \\{y_i \\} = \\{ y_i \in X \mid \langle x, y_i \rangle = \rho_i \ge \rho \\}$$
> > >
> > > The role of this statement is intuitive: a larger approximation factor $c$ corresponds to a larger candidate set $S$, which in turn increases the probability of discovering at least one such neighbor.
> > >
> > > Our rebuttal does not merely patch around this criticism. On the contrary, it removes the questioned assumption altogether and replaces it with a direct localized argument based on the second-moment method. Since removing an assumption strictly strengthens the result, this directly resolves the substance of the concern rather than sidestepping it.
> > >
> > > We therefore respectfully submit that describing this as a “major revision” is disproportionate to the actual scope of the change. The modification is localized to one technical step, does not alter the paper’s main statement or conclusions, and can be completed from the rebuttal sketch with only a few additional details, which we provide below. In our view, the original concern has been resolved in substance.
> > >
> > > ---
> > >
> > > __Notation:__ Let $p = D^{-\frac{1 - \rho}{1 + \rho} + o(1)}$ be the baseline discovery probability for similarity $\rho$, let $p_1 \geq p$ be the average _individual_ discovery probability in $S$ and let $p_2$ be the average _joint_ discovery probability for any _pair_ in $S$.
> > >
> > > __Proof:__ Given indicator variable $X_i$ for each neighbor $y_i$,  where $X_i = 1$ if $y_i \in \text{Top-}m(r_*)$ and $X_i = 0$ otherwise, we have $X = \sum_{i}^{|S|} X_i$ be the total number of neighbors discovered from $S$. Since $X$ is a positive integer, by the Paley-Zygmund inequality, we lower-bound the probability of discovering at least one $y_i \in S$:
> > > $P[X \ge 1] \ge \frac{(E[X])^2}{E[X^2]}$. Then by computing $E[X] = |S| p_1$ and
> > > $E[X^2] = \sum_{i, j} E[X_i X_j] = |S| p_1 + |S| (|S| - 1) p_2$, we can achieve the lower bound stated above.
> > >
> > > $$P[\exists y_i \in S \text{ and } y_i \in \text{Top-}m(r_*)] \ge \frac{|S|^2 p^2_1}{ |S| p_1 + |S| (|S| - 1) p_2} = \frac{p_1^2}{p_2 + (p_1 - p_2) / |S|} .$$

---

### Official Review · Reviewer_4ym4 · 2026-03-12

**Soundness:** 3
**Presentation:** 3
**Significance:** 2
**Originality:** 2
**Overall Recommendation:** 3
**Confidence:** 4

**Summary:**

This paper proposes Random Projection Landmarks (RPL) as an initialization strategy to accelerate the widely used NNDescent algorithm for approximate k-nearest neighbor graph construction, with a focus on settings using cosine similarity. The paper provides rigorous theoretical analysis demonstrating that RPL yields, with high probability, initial neighbor sets containing c-approximate nearest neighbors and improves the probability of discovering true nearest neighbors in subsequent iterations. Extensive empirical results on million-scale, high-dimensional datasets show that RPL significantly accelerates convergence and reduces distance computations relative to existing methods, maintaining or improving final graph recall compared to state-of-the-art initializations.

**Compliance With Llm Reviewing Policy:**

Affirmed.

**Key Questions For Authors:**

Q1 Could you supplement experiments to compare with similar methods?

Q2 Does the performance of your method remain stable on ultra-high-dimensional datasets?

Q3 Does RPL introduce additional memory structures compared with the original NNDescent?

**Limitations:**

Yes

**Strengths And Weaknesses:**

Strengths

S1 The paper addresses a bottleneck in large-scale kNN graph construction and delivers a theoretically grounded, lightweight improvement that is broadly applicable to NNDescent

S2 Theoretical results are carefully worked out, providing clear probabilistic intuition about the “bridge” improvements from RPL initialization, underpinned by detailed lemmas, theorems, and an appendix with full proofs

S3 The results demonstrate RPL’s practical superiority over both random initialization and tree-based (RPT) initialization

Weaknesses
W1 While the integration of random projection landmarks into NNDescent is technically sound and practically effective, the overall conceptual innovation is somewhat limited.The method can be viewed as a structured refinement or enhancement of existing ideas (random projection trees), rather than introducing a fundamentally new algorithmic framework.

W2 Lack of sufficient comparisons with other acceleration methods for NNDescent.

---

> ### Author Rebuttal · Authors · 2026-03-29
>
> We thank your reviews and address your concerns below.
>
> __W1. While the integration of random projection landmarks into NNDescent is technically sound and practically effective, the overall conceptual innovation is somewhat limited.The method can be viewed as a structured refinement or enhancement of existing ideas (random projection trees), rather than introducing a fundamentally new algorithmic framework.__
>
> We agree that our paper does not propose a fundamentally new graph-construction framework. Rather, the contribution is a principled enhancement to the NNDescent pipeline: a new initialization mechanism based on random projection landmarks, together with a theoretical account of why this initialization improves subsequent neighbor-of-neighbor refinement.
>
> We believe this contribution is still important because NNDescent remains a core primitive for approximate k-NN graph construction and continues to attract substantial systems and hardware effort (references by Reviewer Uw2T). This indicates that improving NNDescent is not merely incremental engineering, but an important problem with broad downstream relevance.
>
> Moreover, our target setting is practically meaningful. NNDescent-based implementations such as PyNNDescent are used in widely adopted pipelines for neighbor-graph construction in applications including single-cell analysis [1], UMAP visualization [2], and topic modeling [3]. In these settings, a better initialization can yield consistent improvements without changing the downstream workflow.
>
> - [1] SCANPY: large-scale single-cell gene expression data analysis. Genome Biol. 2018
>
> - [2] UMAP: Uniform Manifold Approximation and Projection for Dimension Reduction, 2018
>
> - [3] https://maartengr.github.io/BERTopic/algorithm/algorithm.html
>
> __W2, Q1. Lack of sufficient comparisons with other acceleration methods for NNDescent__
>
> We thank the reviewer for raising this point. We agree that comparisons to more graph-construction systems would be useful. Our empirical objective here, however, is specifically to evaluate whether the proposed random projection landmark scheme improves initialization for NNDescent-style refinement. For that reason, PyNNDescent is the most relevant primary baseline: it is a strong and widely used NNDescent implementation, and its RPT-based initialization is precisely the closest existing approach to our method.
>
> We also agree that EFANNA [A] is relevant and should be discussed more explicitly. Both EFANNA and PyNNDescent combine a tree-based candidate-generation stage with NNDescent refinement, whereas [B] focuses on tree-based local graph construction without the NNDescent refinement stage. Thus, these methods are closely related in spirit, but PyNNDescent provides the cleanest controlled baseline for isolating the contribution of initialization.
>
> We will revise the paper to clarify that our experiments support a scoped claim, improved initialization within NNDescent pipelines, rather than a comprehensive comparison against every end-to-end k-NN graph construction system.
>
> - [A] Efanna: An extremely fast approximate nearest neighbor search algorithm based on knn graph, 2016
>
> - [B] Scalable k-NN graph construction for visual descriptors, 2012
>
> __Q2. Does the performance of your method remain stable on ultra-high-dimensional datasets?__
>
> We have not yet evaluated RPL on truly ultra-high-dimensional datasets beyond the range $d \in \\{200, 300, 768 \\}$, so we do not want to overclaim stability in that regime. Our expectation is that performance depends more on the effective/intrinsic dimensionality than on the ambient dimension alone: if the data lie near a lower-dimensional structure, the random-projection landmarks and the subsequent NNDescent refinement should remain useful.
>
> The main obstacle to verifying this experimentally is the cost of computing exact ground-truth k-NN graphs for million-point datasets in very large dimensions.
>
> __Q3. Does RPL introduce additional memory structures compared with the original NNDescent?__
>
> Yes. RPL introduces an additional $O(Ln)$ memory cost, where $n$ is the number of data points and $L$ is the number of hash tables. This overhead comes from storing the landmark-bucket assignments used during initialization.
>
> However, this extra memory is  comparable to the auxiliary tree structures already used by PyNNDescent for random-projection-tree initialization. In particular, RPL does not change the asymptotic $O(nk)$ memory required for storing the final approximate k-NN graph itself. It only adds a lightweight initialization structure.

---

### Decision · Program_Chairs · 2026-04-30

**Decision:**

Reject

**Comment:**

The paper proposes an initialization strategy for NNDescent, a popular algorithm for constructing approximate k-nearest neighbor graphs. The key idea is to use random projection directions as "landmarks" to generate better initial neighbor sets, resulting in faster convergence.

Reviewers generally appreciated the importance of the problem and theoretical guarantees. However, they also raised multiple issues. Several reviewers expressed concerns about the paper clarity and the technical correctness. Although the authors provided extensive clarifications during the rebuttal, reviewers believed that implementing those clarifications would require too major a revision of the paper. Some reviewers were also concerned about the limited novelty of the ideas, and insufficient coverage of prior work. Overall, all reviewers recommended rejection.